# CombiMOTS: Combinatorial Multi-Objective Tree Search for Dual-Target Molecule Generation

**Thibaud Southiratn** [1]  **Bonil Koo** [2 3]  **Yijingxiu Lu** [1]  **Sun Kim** [1 2 3 4]

## Abstract

Dual-target molecule generation, which focuses on discovering compounds capable of interacting with two target proteins, has garnered significant attention due to its potential for improving therapeutic efficiency, safety and resistance mitigation. Existing approaches face two critical challenges. First, by simplifying the complex dual-target optimization problem to scalarized combinations of individual objectives, they fail to capture important trade-offs between target engagement and molecular properties. Second, they typically do not integrate synthetic planning into the generative process. This highlights a need for more appropriate objective function design and synthesis-aware methodologies tailored to the dual-target molecule generation task. In this work, we propose CombiMOTS, a Pareto Monte Carlo Tree Search (PMCTS) framework that generates dual-target molecules. CombiMOTS is designed to explore a synthesizable fragment space while employing vectorized optimization constraints to encapsulate target affinity and physicochemical properties. Extensive experiments on real-world databases demonstrate that CombiMOTS produces novel dual-target molecules with high docking scores, enhanced diversity, and balanced pharmacological characteristics, showcasing its potential as a powerful tool for dual-target drug discovery. The code and data is accessible through https://github.com/Tibogoss/CombiMOTS.

[1]Department of Computer Science and Engineering, Seoul National University, Seoul, Republic of Korea [2]Interdisciplinary Program in Bioinformatics, Seoul National University, Seoul, Republic of Korea [3]AIGENDRUG Co., Ltd., Seoul, Republic of Korea [4]Interdisciplinary Program in Artificial Intelligence, Seoul National University, Seoul, Republic of Korea. Correspondence to: Sun Kim <sunkim.bioinfo@snu.ac.kr>.

*Proceedings of the $42^{nd}$ International Conference on Machine Learning*, Vancouver, Canada. PMLR 267, 2025. Copyright 2025 by the author(s).

## 1. Introduction

Complex diseases and disorders such as cancers are generally caused by intricate interacting pathways. To treat them, traditional therapies focused on the single-target paradigm by developing drugs selective towards a unique mechanism, therefore "blind to other processes" yet equally involved in biological systems (Medina-Franco et al., 2013). Single-target drugs are prone to resistance and can risk the activation of unwanted compensatory signaling pathways (Yang et al., 2024b). To overcome this limitation, a rising field of study aims to develop treatments which exhibit better efficacy by interacting with multiple related targets. An intuitive approach known as combination therapy (Kummar et al., 2010; Foucquier & Guedj, 2015) consists in administering various synergetic single-target medications to address distinct but related pathogenic mechanisms. However, these approaches often risk adverse effects or poor treatment adherence (Ye et al., 2023). As promising alternatives, dual-target drugs are increasingly drawing attention, demonstrated by a trend in FDA-approved drugs (Li et al., 2016; Lin et al., 2017; Layman et al., 2024). They involve one agent simultaneously interacting with two targets with better pharmacokinetics and safety profiles, thus avoiding undesirable drug-drug interactions (Hopkins et al., 2006). This inevitably imposes new challenges: (1) The Structure-Activity Relationship (SAR) profile of considered compounds must consider different biological targets (Morphy & Harris, 2012; Raghavendra et al., 2018). (2) The identified compounds have to be synthesizable through known experimental protocols.

In the past few years, deep generative models (Zeng et al., 2022) and geometric deep learning (Powers et al., 2023) tackled SAR for single-target drug design, but the data scarcity and expensive computational cost limit their application on the dual-target setting. Recently, machine learning models allowed to bridge the gap between predictive and intuitive dual-target drug design (Feldmann et al., 2021). Notably, great efforts were made in Fragment-Based Drug Discovery (FBDD) by leveraging property predictors and oracles to extract, combine and autoregressively complete molecular substructures, supposedly responsible for the bioactivity of known compounds on both given targets

(Jin et al., 2020; Lee et al., 2023; Chen et al., 2024). Nevertheless, the key limitation of such methods is the translation from a multi-objective problem to a single-objective formulation. Despite integrating various constraints, their reward function aggregates weighted objectives into a scalar value without considering potential conflicts, leading to overall high-scoring generations with imbalanced properties (Fromer & Coley, 2023; Luukkonen et al., 2023). More specifically, by assuming a convex search space, distinguishing between competing characteristics becomes intractable and resulting molecules can exhibit idealistic properties while being nearly impossible to synthesize (Gao & Coley, 2020; Stanley & Segler, 2023). Furthermore, solely relying on desirability metrics during the generative process cannot guarantee true synthesizability (Vinkers et al., 2003; Ertl & Schuffenhauer, 2009). Consequently, new frameworks also need to condition their processes on synthetic routes (Segler & Waller, 2017; Segler et al., 2018). Swanson et al. (2024) applied Monte-Carlo Tree Search (MCTS) to assemble make-on-demand molecules with associated synthetic procedures yielding an average success rate over 80%, but fall again into the single-objective limitation.

Other fields such as robotics and the computational game domain employ Pareto optimization to explore attainable solutions presenting optimal tradeoffs regardless of competing objectives and property imbalance (Luc, 2008). This flexibility enables to adjust priorities and selection criteria as new information becomes available, without needing to restart the optimization process with different weightings. Few recent works apply Pareto optimality to MCTS in the fields of multi-constrained retrosynthesis planning (Lai et al., 2025) and atom-by-atom molecule SMILES generation (Yang et al., 2024a). However, its application to dual-target molecule generation remains underexplored as the choice of objectives, state space or action space is challenging.

In this paper, we draw inspiration from Pareto MCTS (PMCTS) to propose a novel approach to combinatorial multi-objective drug design. We propose a fragment-based molecule generation approach that optimizes multi-constrained properties of dual-targets within a synthesizable space. To achieve this: (1) We perform search space reduction by extracting target-aware fragments from known target inhibitors and mapping them to industry-available building blocks. (2) We extend PMCTS to a combinatorial framework alongside vectorized objectives tailored for dual-target molecule generation, ultimately generating interpretable compounds exhibiting better tradeoffs in all objectives compared to competing methods. We experimentally validate the effectiveness of our method through the case of dual inhibitor generation as a real world scenario. We summarize our contributions as follows:

- We propose Combinatorial Multi-Objective Tree Search (CombiMOTS), extending MCTS to Pareto optimization and fragment-based drug discovery to narrow down a large molecule fragment space into a synthesizable one, specific to any target pair.
- Through the task of dual inhibitor generation, we demonstrate and analyze CombiMOTS' ability to find optimal candidates across conflicting objectives in complex multi-objective multi-target settings.
- In particular, we evaluate our model on the GSK3$\beta$-JNK3 pair and curate data for two new target pairs, EGFR-MET and PIK3CA-mTOR to illustrate the utility of CombiMOTS in dual-target drug design.
- On all experiments, CombiMOTS consistently generates molecules that yield better diversity and trade-off between docking score, drug-likeness, and synthetic accessibility.

## 2. Related Work

**Multi-Objective Molecule Generation** The success of prior generative models for single-target/objective drug design does not trivially apply to multi-objective parameterization (Angelo et al., 2023). These methods commonly optimize embeddings of molecular sequences or graphs, relying on bayesian inference or reinforcement learning. Multi-objective molecule generation approaches can be categorized into three categories. The first category uses encoder-decoder architectures (Jin et al., 2018; 2020) or autoregressive frameworks (Gong et al., 2024; Yang et al., 2024a) to attempt learning continuous latent representations of target compounds to generate candidates with encouraging predicted properties. However, they highly depend on the quality of the learnt latent space, which is burdensome in multi-objective tasks where data is scarce. The second category being reinforcement learning, operates in the explicit chemical space to allow more robust training, but are hard to train due to reward sparsity when combining multiple objectives (Guimaraes et al., 2017; Olivecrona et al., 2017; Popova et al., 2018). More specific to dual-target drug design, Li et al. (2018); Jin et al. (2020); Xie et al. (2021) utilize a machine learning property predictor to guide the generation of bio-active compounds, but still require high-quality labeled data to properly work. As the third category, Structure-Based Drug Design (SBDD) methods integrate binding affinity indicators to moderate data scarcity while enhancing SAR with target proteins (Chen et al., 2024; Zhou et al., 2024). Compared to these methods, we utilize both property predictors and binding score oracles to cooperatively account for pharmacological characteristics and dual-target affinity.

**Fragment-Based Molecule Generation** Fragment-based approaches have emerged as a promising direction in drug discovery. The intuition is to derive multi-property candi-

dates from initial single-property molecular substructures as building blocks. Recent research in this area can be broadly categorized into fragment extraction and assembly methods for generation. For fragment extraction, early approaches relied on rule-based bond breaking selection (Yang et al., 2021) or motif frequency-based selection (Kong et al., 2022). The drawback of these heuristic methods generally is to not consider target molecular properties. This motivated succeeding works to try identifying property-aware substructures. Jin et al. (2020); Chen et al. (2024) extract and assemble core fragments from full molecules rather than true fragments, leading to limited novelty and diversity in generation. For generation methods, MCMC sampling strategy can be applied for its flexibility in adding and deleting fragments (Xie et al., 2021), while reinforcement learning (Tan et al., 2022; Du et al., 2023), and VAE-based models for fragment assembly (Maziarz et al., 2021) remain widely used. Interestingly, Swanson et al. (2024) proposed a method that does not require to extract fragments as building blocks. They use the industry-ready Enamine REadily AccessibLe (REAL) Space database (Grygorenko et al., 2020), with building blocks and reaction templates as inputs to MCTS, with a single property predictor guiding the space navigation modeled as chemical reactions. This presents the advantage of keeping a high chance to synthesize identified leads in real world laboratory settings. We propose to extend their work to the dual-target molecule generation task by leveraging property-aware available building blocks, subsequently assembled using a vectorized multi-objective parametrization guided by oracles tailored to the target proteins.

## 3. Preliminary - Pareto Optimization

Multi-objective problems involve optimizing several objectives, often competing. A common approach is to aggregate them into one scalarized reward, naturally inducing oversimplification of the problem's complexity by depending on importance weighting and its *a priori* knowledge. Pareto optimization (El-Sharkawi, 2005) alleviates this issue by maintaining a vectorized problem formulation, thus defining sets of solutions whose components are all considered equal, allowing proper optimization in the attainable solution space. Without loss of generality, we assume a maximization problem over $D \in \mathbb{R}$ objectives, where each solution is characterized by a $D$-dimensional property vector.

**Definition 3.1** (Pareto Dominance)**.** Let $\mathbf{X}$ and $\mathbf{Y}$ be two distinct objects, each associated with their respective objective vectors $\mathbf{P}_X, \mathbf{P}_Y \in \mathbb{R}^D$, where $P_{X,d}$ and $P_{Y,d}$ denote the values of X and Y on the $d$-th objective, respectively. We define Pareto dominance as follows: $X$ dominates $Y$, denoted by $X \succ Y$ (or equivalently $Y \prec X$), if and only if:

i) No coordinate of $\mathbf{P}_X$ is smaller than the corresponding element of $\mathbf{P}_Y$:

$$\forall d \in \{1, 2, \ldots, D\}, P_{X,d} \geq P_{Y,d} . \qquad (1)$$

ii) At least one coordinate of $\mathbf{P}_X$ is larger than the corresponding element in $\mathbf{P}_Y$:

$$\exists d \in \{1, 2, \ldots, D\} \text{ s.t. } P_{X,d} > P_{Y,d} . \qquad (2)$$

If only the first condition is satisfied, $X$ is said to weakly-dominate $Y$, denoted by $X \succeq Y$ or $Y \preceq X$. When neither $X \succeq Y$ nor $Y \succeq X$ holds, $X$ and $Y$ are said incomparable ($X \| Y$).

*Remark* 3.2. i) and ii) can be summarized as: $X \succ Y$ if $X$ is "never worse" and "at least better than $Y$ in one objective".

**Definition 3.3** (Pareto Fronts)**.** Given an non-empty set of vectors $\mathcal{S} \subset \mathbb{R}^D$ obtained from candidate solutions found in the objective space, the first Pareto front (or Pareto optimal set) consists of all non-dominated solutions:

$$\mathcal{S}_1 = \{\mathbf{P}_X \in \mathcal{S} : \nexists \mathbf{P}_Y \in \mathcal{S} \text{ s.t. } Y \succ X\} . \qquad (3)$$

Subsequent Pareto fronts $\mathcal{S}_k$ are defined recursively by excluding all solutions belonging to the preceding $\{S_i\}_{i=1}^{k-1}$:

$$\mathcal{S}_k = \{\mathbf{P}_X \in \mathcal{S} : \nexists \mathbf{P}_Y \in \mathcal{S} \setminus \bigcup_{i=1}^{k-1} \mathcal{S}_i \text{ s.t. } Y \succ X\} . \qquad (4)$$

*Remark* 3.4. All solutions from the same Pareto front are considered equivalent, as they are either identical or incomparable:

$$\forall \mathcal{S}_k \subset \mathcal{S}, \forall (X, Y) \in \mathcal{S}_k, (X \| Y) . \qquad (5)$$

## 4. Proposed Approach: CombiMOTS

We propose CombiMOTS, a PMCTS method designed to navigate a dual-target specific synthesizable space by leveraging the strengths of both FBDD and Pareto optimization. We first outline in Sec.4.1 how to reduce a large chemical space to available dual-target-specific building blocks. We then describe our reaction-based PMCTS implementation in Sec.4.2. We provide additional implementation details in Appendix D.3.

### 4.1. Synthesizable Search Space Reduction

The estimated size if the drug-like chemical space exceeds $10^{33}$ molecules, making exhaustive search for compounds with desirable chemical and structural properties computationally infeasible (Polishchuk et al., 2013).

To address this challenge, we narrow the search space using curated subsets of Enamine REAL Space (Grygorenko et al.,

2020), refined by Swanson et al. (2024). This ensures that our method remains computationally tractable while maintaining its practical relevance. Given a target pair, we reduce the search space by (1) identifying target-aware fragments from known active inhibitors, and (2) aligning them with the REAL Space building blocks and reactions (Figure 1).

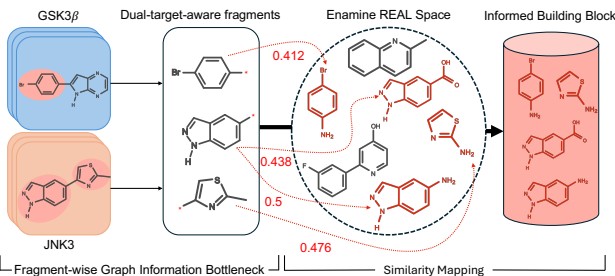

*Figure 1.* Search Space Reduction. Property-aware fragments are extracted for each target using Fragment-wise Graph Information Bottleneck (Lee et al., 2023). By applying a Tanimoto similarity threshold to a synthesizable search space, we curate a final set of informed industry-ready building blocks.

**Fragment Extraction**    To extract chemically meaningful fragments from active molecules, we employ Fragment-wise Graph Information Bottleneck (FGIB) (Lee et al., 2023). FGIB provides a sophisticated approach to extracting substructures using BRICS decomposition (Degen et al., 2008) from known active compounds, highly contributing to their target property. Its theoretical foundation lies in the Graph Information Bottleneck (GIB) principle (Wu et al., 2020), where the objective function balances between preserving property-predictive information and compressing an original molecular graph representation. This is achieved through a dual optimization process, where the goal is to simultaneously maximize information between extracted fragments and target properties, while minimizing the mutual information between fragments and the original graph. Given a molecule graph $G$ and its target property $Y$, GIB identifies an informative set of fragments $G^{\text{sub}}$ by optimizing:

$$\min_{G^{\text{sub}}} - I(G^{\text{sub}}, Y) + \beta I(G^{\text{sub}}, G) . \quad (6)$$

where $I(.|.)$ denotes mutual information and $\beta$ is a positive hyperparameter balancing informativeness and compression.

FGIB utilizes Message Passing Neural Networks (MPNN) to compute graph embeddings and introduces a novel noise injection mechanism that modulates information flow based on fragment importance. This mechanism enhances our model's ability to identify chemically meaningful substructures by selectively preserving informative signals, while reducing redundancy.

**Search Space Reduction**    To ensure interpretable and realistic generations, the core implementation of CombiMOTS relies on predefined building blocks from the Enamine REAL Space. After obtaining dual-target-aware fragments, we curate the final set of building blocks by mapping learnt fragments to existing Enamine building blocks with a Tanimoto similarity above a fixed threshold (Figure 1). We empirically find in Section 5.7.2 that using ECFP4 Morgan fingerprints and a threshold value of 0.4 generally leads to an appropriate search space size ($\sim$14k blocks and $\sim$25M possible reaction products).

### 4.2. Pareto Monte-Carlo Tree Search

#### 4.2.1. MONTE-CARLO TREE SEARCH

MCTS is a decision-making algorithm that systematically explores a search space by balancing exploration and exploitation through four distinct phases. (i) The **selection** phase aims at traversing a search tree from the root node to a leaf node, using the Upper Confidence Bound (UCB) formula to greedily select the most promising actions based on both their estimated UCB value and the uncertainty of those estimates. Given a synthesis tree $T$ to search, a node $n$ with parent $p$, we denote $C$ a positive exploration constant, $R(n)$ the cumulative reward of $n$ and $N(n)$ the number of times $n$ was visited:

$$\text{UCB}(n) = \underbrace{\frac{R(n)}{N(n)}}_{\text{exploitation}} + C \underbrace{\sqrt{\frac{\ln(N(p))}{N(n)}}}_{\text{exploration}} . \quad (7)$$

The 'exploitation' term favors selecting high-property nodes, while the 'exploration' term encourages selecting less explored nodes, jointly guiding the traversal of both promising and uninvestigated outcomes. (ii) In the **expansion** phase, reaching a leaf node triggers the creation of child nodes, each representing a new potential state or action in the search space. (iii) The **simulation** (rollout) phase iteratively applies steps (i) and (ii) from the newly expanded node until reaching a terminal state or predefined depth. (iv) In the **backpropagation** phase, the statistics of all nodes along the traversed path are updated, propagating simulation outcomes upward to refine value estimates and visit counts. This iterative process continues until computational constraints or a convergence criterion are met, progressively constructing a search tree that balances exploration and exploitation while ensuring sufficient search space coverage.

#### 4.2.2. PARETO MCTS FOR DUAL-TARGET MOLECULES

We propose an extended version of MCTS that incorporates Pareto optimization through vectorized properties, integrating property predictors and docking oracles based on combinatorial chemistry principles (Liu et al., 2017) to identify

synthetically accessible compounds that jointly target two proteins.

---

**Algorithm 1** High-Level CombiMOTS Algorithm

---

1: **Input:** Reactions $R$, rollouts $n_{rollout}$
**Require:** Multi-objective $Oracles$
2: Initialize MCTS tree $T$ with empty root node
3: **for** $i = 1$ **to** $n_{rollout}$ **do**
4:    $v \leftarrow \text{Rollout}(T.root)$
5: **end for**
6: **return** Pareto optimal molecules from visited nodes
7:
8: **function** Rollout($node$)
9:    **if** $node$ is a product **then**
10:      **return** $Oracles(node)$ ▷ Backpropagation
11:    **end if**
12:    $children \leftarrow \text{GetChildNodes}(node)$ ▷ Expansion
13:    $selected \leftarrow \underline{\text{ParetoSelection}}(children)$
14:    $v \leftarrow \text{Rollout}(selected)$
15:    **if** $selected$ is a product **then**
16:      $v \leftarrow \text{Elem-wise-max}(v, selected.P)$ ▷ Backprop.
17:    **end if**
18:    Update statistics of $selected$
19:    **return** $v$
20: **end function**

---

The major modification from the single-objective version MCTS described in Section 4.2.1 is the design of multi-dimensional property vectors, enabling element-wise optimization. This formulation allows the application of Pareto principles (Section 3) by optimizing vector coordinates across the full objective space, but it also requires modifications of the MCTS core steps. In this setting, tree traversal can no longer greedily select the highest-scoring node. Given the property vector $\vec{Ora}(n)$ obtained from $Oracles$ and $D$ the number of objectives, Equation (7) becomes the Pareto UCB formula (PUCB):

$$\overrightarrow{PUCB}(n) = \frac{\vec{R}(n)}{N(n)} + C \times \overrightarrow{Ora(n)} \sqrt{\frac{\ln(D) + 4 \times \ln(1 + N(p))}{1 + N(n)}}.$$

(8)

The Figure 2 and Algorithm 1 illustrate a complete rollout of CombiMOTS, while Algorithm 2 provides a detailed description of the algorithm. During the selection step, after computing the PUCB vector of all child nodes, the non-dominated nodes form a local Pareto front, from which the next selected node is randomly sampled. In our problem, each node represents a tuple of multiple building blocks. Upon reaching a leaf node, the expansion step generates new child nodes in two ways: (a) by applying all possible reactions to the building blocks in the node and creating one node per product, or (b) incorporating all matching reactants

of each building block and creating one node per addition. The rollout iteratively repeats these steps until a reaction product is reached (terminal node). When a terminal node is reached, the backpropagation step updates the reward vectors of all nodes along the traversal path, before initiating the next rollout. The algorithm ends once the specified number of rollouts is completed, returning all encountered reaction products.

## 5. Experiments

### 5.1. Datasets

As our work focuses on dual-target molecule generation, we demonstrate the practical utility of CombiMOTS in real-world scenarios by evaluating it on three disease-related protein target-pairs. Notably, we curate and release new datasets for the EGFR-MET and PIK3CA-mTOR pairs. Detailed data statistics and curation methods are provided in Appendices D.1 and M.

**GSK3$\beta$-JNK3** These are two kinases related to Alzeihmer's Disease (AD) (McCubrey et al., 2014; Koch et al., 2015). Designing dual-inhibitors for these targets open possibilities towards more efficient AD therapies. We utilize the data curated by Li et al. (2018) as a commonly used benchmark for dual-target generation (Jin et al., 2020; Xie et al., 2021; Chen et al., 2024).

**EGFR-MET** In non-small cell lung cancer (NSCLC), resistance to EGFR inhibitors often involves compensatory activation of MET, leading to tumor progression (Fang et al., 2024). Therefore, a dual-target molecule that simultaneously inhibits EGFR and MET could overcome drug resistance and improve therapeutic outcomes (Ren et al., 2024). This strategy is supported by multiple studies showing the potential benefits of combined EGFR and MET inhibition in patients with EGFR mutations and MET amplification or overexpression (Fang et al., 2024; Ren et al., 2024).

**PIK3CA-mTOR** Mutations in PI3K, particularly PIK3CA, and dysregulation of mTOR contribute to cancer cell growth, proliferation, and survival (Fruman & Rommel, 2014). Dual inhibitors targeting both PI3K and mTOR provide a more comprehensive blockade of this pathway, addressing potential resistance mechanisms, as demonstrated by the PI3K/mTOR dual inhibitor VS-5584 (Hart et al., 2013). Simultaneous inhibition of both targets is expected to more effectively suppress PI3K-mTOR signaling compared to isoform-selective inhibitors, potentially overcoming feedback activation that limits single-target therapies (Rodrik-Outmezguine et al., 2016).

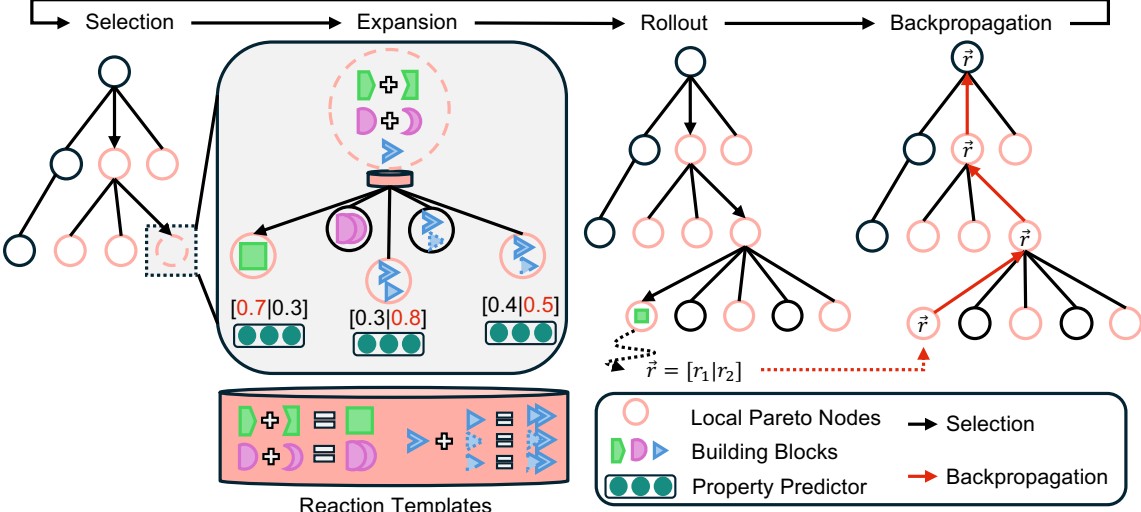

*Figure 2.* One full iteration of the CombiMOTS Algorithm. (i) Selection from local Pareto fronts occurs until a leaf node to expand is found. (ii) During expansion, two types of nodes are created by consulting the reaction templates: those representing products from current building blocks and those incorporating compatible reactants. Upon creation, oracles predict properties for all child nodes to establish their local Pareto front. We illustrate property vectors with two objectives. (iii) Selection and Expansion steps iteratively occur until a reaction product is found. (iv) The property vector of the reaction product is backpropagated up the path.

## 5.2. Baselines

**RationaleRL** (Jin et al., 2020) identifies important structural components from active compounds through MCTS. It then systematically merges these components while maintaining their shared structural elements, followed by targeted fine-tuning of a generative model to append appropriate side chains. **REINVENT** (v3.2) (Blaschke et al., 2020) adopts a sequence-based approach, utilizing a recurrent neural network architecture to generate molecular SMILES strings. The framework incorporates reinforcement learning techniques to optimize multiple molecular properties simultaneously, enabling the design of compounds with activity across multiple objectives. **MARS** (Xie et al., 2021) selects the top 1,000 fragments from the ChEMBL dataset based on their size and frequency. The obtained fragments have less than 10 heavy atoms with frequent occurrence in the dataset. It employs graph neural networks (GNNs) and Markov Chain Monte Carlo (MCMC) sampling to refine molecules with enhanced properties, from addition or deletion of these fragments.

## 5.3. Evaluation Metrics

For each model, we generate $N = 10,000$ molecules for all target pairs, then compare the performance of models to assess their ability to generate original and biochemically potent compounds. We evaluate generations with metrics fitting two categories:

**Originality** We gauge the models' ability to generate original compounds using the following metrics. **Validity (%)**:

The proportion of molecules adhering to all chemical rules. **Uniqueness (%)**: The proportion of distinct molecules within the generated set, where a low value suggests repetitive generation. **Novelty (%)**: As defined by Olivecrona et al. (2017), the proportion of generated molecules whose nearest training set dual-active neighbor has a Tanimoto similarity score below 0.4, reflecting the model's capacity to generate sufficiently different, yet desirable compounds. **Diversity (%)**: The proportion of generated molecules with a pairwise Tanimoto similarity below 0.4 across all generations, indicating the model's ability to explore broad regions of the chemical space rather than a narrow subset. **Activity Success Rate (%)**: The proportion of generated molecules predicted to be active towards both target proteins, indicating the model's ability to identify potential dual-inhibitors. We denote such molecules as "dual-actives".

**Quality** We interpret property distributions by visualizing density plots of dual-active compounds. We analyze: **Docking Score** (DS) as an approximation of binding affinity between target proteins and molecule ligands. A lower docking score indicates better molecular interactions. **Quantitative Estimate of Drug-likeness** (QED) quantifies the potential of a compound presenting desirable drug capabilities based on properties such as topological polar surface area and the number of hydrogen donors/acceptors. Ranging from 0 to 1, a higher score is preferable. **Synthetic Accessibility score** (SA) measures how difficult a given compound is to synthesize, based on fragment contribution and complexity penalty. Ranging from 1 to 10, compounds with a low SA score are easier to obtain.

Details on used oracles and molecular docking settings are given in Appendix D.3.

## 5.4. Objectives

RationaleRL and MARS are reproduced in their best reported settings. Following Li et al. (2018), they use random forest (RF) classifiers for each target, alongside QED and SA scores to condition their generation. REINVENT is reproduced with the same property predictors with QED score only as SA score is not supported.

The objectives of CombiMOTS integrate both biological activity and structural constraints. Following Li et al. (2018); Jin et al. (2020); Xie et al. (2021); Swanson et al. (2024), we use activity towards both targets as an empirical estimator of inhibition, derived from real experimental assays used to train machine learning predictors. For instance, the GSK3$\beta$-JNK3 training data originates from PubChem and ChEMBL (Gaulton et al., 2012; Kim et al., 2016), filtered based on bioactivity data, particularly IC$_{50}$ values. We employ a Chemprop (D-MPNN) classifier for multi-task prediction of dual-target activity. Unlike baselines, we prioritize molecular docking scores for both targets over QED and SA. Specifically for dual-inhibitor design, we argue that to accurately consider SAR during generation, "We Should at Least Be Able to Design Molecules That Dock Well" (Cieplinski et al., 2020): CombiMOTS successfully captures ligand-binding affinity, as showcased in Section 6.1. Details and justification on objective design are given in Appendices D.3, E.1.2 and E.1.3.

We conduct additional experiments in Appendix E. They include comparative analysis of: (i) A scalarized version of CombiMOTS to emphasize the utility of Pareto MCTS. (ii) A variant prioritizing QED/SA over docking scores and a six objective version which also incorporates QED and SA scores. (iii) Implementations of MARS and REINVENT aligned with our objectives by replacing QED and SA with docking score. (iv) A comparison against AIxFuse (Chen et al., 2024), a model using collaborative reinforcement learning and active learning, on the DHODH-ROR$\gamma$t pair to investigate how our model handles data scarcity.

## 5.5. Experimental Settings

For RationaleRL, REINVENT and MARS, we generate N = 10,000 molecules across all three target pairs. As CombiMOTS is fundamentally a search method, it does not 'generate' a fixed number of outputs but instead returns all reaction products within a computational budget. We fix $n_{rollout}$ = 50,000 for all tasks (GSK3$\beta$-JNK3, EGFR-MET, PIK3CA-mTOR) and randomly sample 10,000 found dual-active molecules. *Therefore, we do not need to compute activity success rate for CombiMOTS.*

## 5.6. Results

As shown in Figures 3 and 7, CombiMOTS consistently identifies optimal tradeoffs across all three target pairs under each metric. In contrast, competing methods exhibit inconsistent performance across target pairs, often generating compounds with suboptimal properties or excelling in a single objective while underperforming in others. Notably, displayed density distributions are normalized, highlighting the rarity of high-quality candidates produced by alternative methods. Furthermore, Table 1 suggests that REINVENT and RationaleRL can generate invalid SMILES, and all models struggle to generate unique, novel and diverse compounds on all tasks. In every setting, CombiMOTS outperforms other methods in generating novel and diverse molecules with well-balanced physicochemical and structural properties, as visualized in Figure 4. Finally, we compute the respective first Pareto front of all models and report in Table 2 their average R2-distance to the utopia point, as well as the size of their optimal set (details in Appendix D.3): our approach steadily finds more Pareto optimal solutions. This evidence demonstrates the effectiveness of our approach in navigating the complex multi-objective landscape of dual-target drug design.

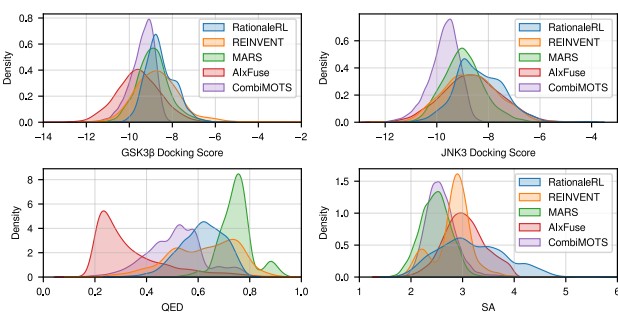

*Figure 3.* Normalized distributions of the generated molecules across dual-docking scores, QED and SA scores on the GSK3$\beta$-JNK3 target pair.

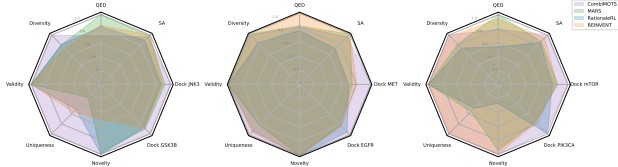

*Figure 4.* Radar charts summarizing performance on the GSK3$\beta$-JNK3 (left), EGFR-MET (middle) and PIK3CA-mTOR (right) tasks. We report average values of originality metrics and median values of quality metrics, normalized to the best score on each axis.

## 5.7. Ablation Studies

To demonstrate the effectiveness of the Search Space Reduction step, we conduct experiments to emphasize the role of FGIB in fragment extraction (Section 5.7.1) and the impact

*Table 1.* Performance comparison across different target pairs and molecular generation models based on 10,000 sampled molecules from three runs. (**: $p < 0.001$; *: $p < 0.01$)

| Target Pairs | Models | Metrics | | | | |
| --- | --- | --- | --- | --- | --- | --- |
| | | Valid. (%) | Uniq. (%) | Novel. (%) | Div. (%) | Act. SR (%) |
| GSK3$\beta$-JNK3 | RationaleRL | 99.95$_{\pm0.01}$ | 50.46$_{\pm0.05}$ | 100$_{\pm0.00}$ | 70.92**$_{\pm0.01}$ | 99.97$_{\pm0.01}$ |
| | REINVENT | 99.86$_{\pm0.06}$ | 56.14$_{\pm0.77}$ | 49.72$_{\pm0.37}$ | 64.52**$_{\pm0.19}$ | 99.33$_{\pm0.15}$ |
| | MARS | 100$_{\pm0.00}$ | 26.02$_{\pm3.04}$ | 97.87$_{\pm0.56}$ | 70.64$_{\pm2.85}$ | 99.86$_{\pm0.10}$ |
| | CombiMOTS | 100$_{\pm0.00}$ | 100$_{\pm0.00}$ | 99.96$_{\pm0.01}$ | 88.67$_{\pm0.52}$ | - |
| EGFR-MET | RationaleRL | 99.95$_{\pm0.02}$ | 92.22$_{\pm0.21}$ | 100$_{\pm0.00}$ | 75.78$_{\pm0.04}$ | 99.81$_{\pm0.01}$ |
| | REINVENT | 99.15$_{\pm1.55}$ | 94.82$_{\pm0.14}$ | 99.80$_{\pm0.05}$ | 89.59$_{\pm0.04}$ | 70.47$_{\pm0.12}$ |
| | MARS | 100$_{\pm0.00}$ | 80.30$_{\pm0.88}$ | 99.82$_{\pm0.08}$ | 91.89*$_{\pm0.10}$ | 69.04$_{\pm0.53}$ |
| | CombiMOTS | 100$_{\pm0.00}$ | 100$_{\pm0.00}$ | 99.81$_{\pm1.62}$ | 90.75$_{\pm0.40}$ | - |
| PIK3CA-mTOR | RationaleRL | 99.99$_{\pm0.01}$ | 48.34$_{\pm0.33}$ | 27.04$_{\pm0.39}$ | 66.55$_{\pm0.04}$ | 99.98$_{\pm0.02}$ |
| | REINVENT | 99.47$_{\pm0.08}$ | 98.39$_{\pm0.09}$ | 99.96$_{\pm0.01}$ | 87.71*$_{\pm0.04}$ | 97.60$_{\pm0.20}$ |
| | MARS | 100$_{\pm0.00}$ | 55.00$_{\pm7.78}$ | 94.42$_{\pm3.21}$ | 73.82$_{\pm2.12}$ | 99.86$_{\pm0.23}$ |
| | CombiMOTS | 100$_{\pm0.00}$ | 100$_{\pm0.00}$ | 99.99$_{\pm0.01}$ | 90.88$_{\pm0.36}$ | - |

*Table 2.* Average R2-distance to the utopia point normalized in range 0 to 2 (lower is better) and size of Pareto optimal sets.

| Model/Task | GSK3$\beta$-JNK3 | EGFR-MET | PIK3CA-mTOR |
| --- | --- | --- | --- |
| CombiMOTS | 0.8075 (**174**) | **0.8419 (268)** | 0.8143 (**210**) |
| MARS | **0.7830** (102) | 0.8845 (142) | 0.8144 (89) |
| RationaleRL | 0.8307 (137) | 0.9388 (148) | 0.8852 (113) |
| REINVENT | 0.8223 (93) | 0.8518 (177) | **0.8035** (174) |

of Tanimoto similarity thresholds on the search space size (Section 5.7.2).

### 5.7.1. DUAL-TARGET-AWARE FRAGMENT EXTRACTION

FGIB extends BRICS by breaking molecules into retrosynthetically interesting fragments, which is preferred over rule/frequency-based methods when combined with Enamine REAL Space. Some works adapt BRICS to needs with a fixed goal e.g. pBRICS (Vangala et al., 2023) for ADMET explanability, thus not suited for the dual-inhibition task where target proteins are user-defined. Our goal is to capture high-property fragments for any target property. To further justify our choice, we report results from 10k rollouts on the GSK3$\beta$-JNK3 task when replacing FGIB with naive BRICS and MiCaM (Geng et al., 2023), a connection-aware motif-mining approach to decompose known active compounds. Table 3 shows that FGIB successfully identified goal-specific fragments, resulting in a significantly higher rate of dual-active compounds. For other metrics (Appendix B), all methods exhibit similar performance, which is sound as FGIB only impacts the search space through the selection of initial blocks: MCTS objectives and convergence are "as good" but the attainable space contains more compounds likely exhibiting dual-activity potential.

### 5.7.2. SENSITIVITY OF THE SEARCH SPACE SIZE

We investigate the use of various thresholds and report results from 10k rollouts on the GSK3$\beta$-JNK3 task for lower and higher values of 0.3, 0.5 & 0.6. Table 4 shows that (i) the threshold greatly affects the search space size due to the small nature of FGIB fragments and Enamine blocks. Tani-

*Table 3.* Search space size and dual-actives candidates over all generations using FGIB, Naive BRICS and MiCaM. $\neq$ *refers to blocks not used in the specified method.*

| Method | #Building Blocks | #Possible Products | #Dual-Actives |
| --- | --- | --- | --- |
| BRICS | 4,430 (747 $\neq$ FGIB) | $\sim$1.7M | 1,815/12,559 (14.45%) |
| MiCaM | 12,274 (8,428 $\neq$ FGIB) | $\sim$26M | 1,445/13,263 (10.90%) |
| **FGIB (Base)** | 14,366 (10,683 $\neq$ BRICS, 10,520 $\neq$ MiCaM) | $\sim$25M | 3,662/15,423 (**23.74%**) |

moto metric being fingerprint-based, changing a small motif is relatively impactful. (ii) For higher thresholds, the search space is too small to allow good exploration. There is a significant drop in number of dual-actives, and other metrics (Appendix B) suggest that thresholds above 0.6 yield a drop in diversity (88.67→78.73%) and consistency across molecular properties. (iii) For lower thresholds, CombiMOTS has "more room to explore" but finds worse tradeoffs across QED and less dual-actives. As more reactions are possible, Pareto fronts are larger and convergence to optimal solutions is slower (Appendix L). Optimal thresholds are specific to the target properties, but our experiments suggest that a search space of magnitude 10M yields better convergence within a reasonable budget ($\sim$10k rollouts).

*Table 4.* Search space size and dual-actives candidates over all generations using different similarity threshold values.

| Threshold | #Building Blocks | # Possible Products | # Dual-Actives |
| --- | --- | --- | --- |
| 0 (Full Space) | 139,493 | >31B | (Not Performed) |
| 0.3 | 54,811 | $\sim$478M | 2,833/13,556 (20.90%) |
| 0.4 (Base) | 14,366 | $\sim$25M | 3,662/15,423 (**23.74%**) |
| 0.5 | 3,737 | $\sim$1.1M | 1,380/11,632 (11.86%) |
| 0.6 | 858 | $\sim$43k | 317/9,433 (3.36%) |

## 6. Case Studies

### 6.1. Visualization of potential dual-inhibitors for GSK3$\beta$-JNK3

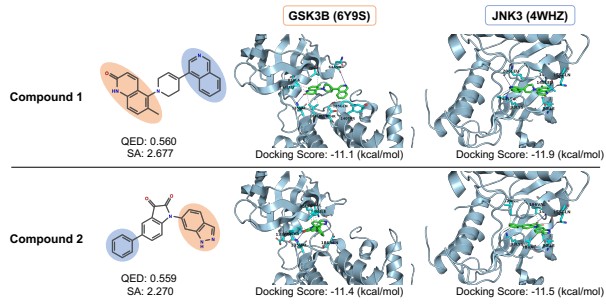

*Figure 5.* Examples of molecules generated by CombiMOTS on the GSK3$\beta$-JNK3 target pair. They both yield high drug-likeness and synthesizability metrics, and different (colored) parts of the compounds are positioned into each protein pocket.

We show in Figure 5 two example dual-target molecules

(Compound 1 and Compound 2) generated by CombiMOTS to inhibit both GSK3$\beta$ and JNK3. Through *in silico* docking, each compound was predicted to occupy the ATP binding pocket of both targets, reproducing key interaction patterns highlighted in previous structural and biochemical studies (Lu et al., 2018; Cheng et al., 2022). Notably, hydrogen-bond interactions in the hinge region (e.g., *VAL135* in GSK3$\beta$; *MET149* in JNK3) and the engagement of hydrophobic pockets (*VAL70, LEU132* in GSK3$\beta$; *ILE70, VAL78* in JNK3) were consistently observed, indicating strong binding complementarity.

Compound 1 exhibited strong predicted affinity, with docking scores of –11.1 kcal/mol for GSK3$\beta$ and –11.9 kcal/mol for JNK3. Hydrophobic interactions with residues such as *VAL70* (GSK3$\beta$) and *ILE70* (JNK3) stabilized its positioning in catalytic clefts. Hydrogen bonds with residues such as *ASN64* (GSK3$\beta$) and *LYS93* (JNK3) further reinforced dual-target binding, aligning with known hinge-directed inhibitor interactions (Quesada-Romero et al., 2014; Cheng et al., 2022).

Similarly, Compound 2 achieved favorable docking scores (–11.4 kcal/mol for GSK3$\beta$, –11.5 kcal/mol for JNK3). Key hydrogen bonds with *ASP133* (GSK3$\beta$) and *MET149* (JNK3), along with hydrophobic contacts involving *ILE62* (GSK3$\beta$) and *ILE70* (JNK3), support its effective active-site engagement (Quesada-Romero et al., 2014; Lu et al., 2018; Cheng et al., 2022). These interactions indicate robust dual-target binding for both compounds. The full list of identified interactions is found in Appendix G.

Beyond binding predictions, both compounds display QED values around 0.56 and SA scores under 3, suggesting a reasonable balance of drug-like properties and ease of synthesis. These findings highlight the promise of CombiMOTS-driven design in generating dual-target molecules that not only recapitulate essential interaction motifs but also meet fundamental criteria for drug discovery.

### 6.2. Toxicity Prediction

To further demonstrate the applicability of our method in drug discovery, we estimate the toxicity profile of molecules generated by CombiMOTS. Following Swanson et al. (2024), we train an ensemble of ten Chemprop models on the ClinTox dataset (Wu et al., 2018) using clinical toxicity binary labels. The dataset comprises 1,478 molecules, with 1,366 experimentally validated as non-toxic (0) and 112 identified as toxic (1).

Figure 6 compares the predicted toxicities of 10,000 CombiMOTS-generated molecules for the GSK3$\beta$-JNK3 task against ground truth compounds from ClinTox. While some generations tend to be toxic, we make two key observations: (i) The average predicted toxicity of 10,000 samples

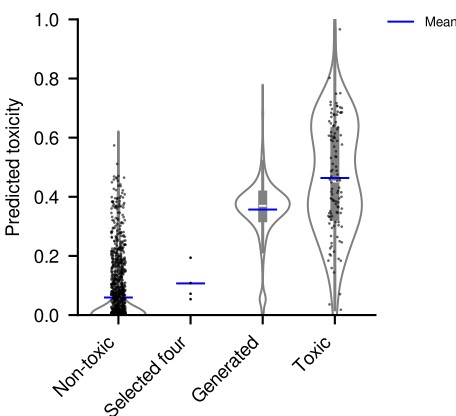

*Figure 6.* Toxicity predictions for generated molecules on the GSK3$\beta$-JNK3 target pair. 'Generated' refers to all 10,000 generated samples. 'Selected four' refers to the compounds discussed in 6.2.

remains below that of the 112 truly toxic compounds, with some generations predicted as potentially non-toxic. (ii) Four randomly selected samples with predicted toxicity below 0.2 exhibit high scores across all properties (Figure 19). Despite toxicity not being optimized, these four samples appear in the first two Pareto fronts, suggesting that Pareto-optimal solutions may hold promise for lead identification. We perform an additional experiment on toxicity optimization in Appendix I.2.

## 7. Conclusion

We proposed CombiMOTS, a novel approach to combinatorial multi-objective drug design by integrating PMCTS with fragment-based molecule generation. By leveraging dual-target-aware fragments available in industry and extending PMCTS to a combinatorial framework, our method effectively navigates the complex design space of dual-target drug discovery while ensuring synthesizability. Through comprehensive evaluations on multiple dual-target inhibitor tasks, we demonstrate that CombiMOTS consistently outperforms existing methods by generating diverse and interpretable molecules that achieve favorable trade-offs across key objectives. Our results highlight the potential of CombiMOTS as a valuable tool in rational drug design, paving the way for further advancements in multi-objective optimization for molecular generation.

Future directions could extend our framework to larger chemical libraries e.g., Freedom Space (Protopopov et al., 2024) or broader multi-objective applications. While Appendix I offers preliminary investigations of selective molecule generation, toxicity optimization and protein-protein interaction modulation, we envision future works to explore various strategies to expand the applicability of CombiMOTS in real-world scenarios.

## Acknowledgements

This study was supported by the Bio & Medical Technology Development Program of the National Research Foundation (NRF), funded by the Ministry of Science and ICT, Republic of Korea (RS-2022-NR067933), the NRF grant funded by the Korea government (MSIT, RS-2023-00257479), the Korea Health Technology R&D Project through the Korea Health Industry Development Institute (KHIDI), funded by the Ministry of Health & Welfare, Republic of Korea (RS-2024-00403375), and Institute of Information & communications Technology Planning & Evaluation (IITP) grant funded by the Korea government (MSIT, RS-2021-II211343, Artificial Intelligence Graduate School Program (Seoul National University). This study is also funded by AIGEN-DRUG Co., Ltd. and ICT at Seoul National University provided research facilities.

## Impact Statement

We release CombiMOTS to be effective in real-world drug discovery tasks and useful to practitioners. Common risks involve using this tool to design harmful compounds or having complete faith in machine-learned predicted properties with no *post-hoc* validation. For instance, metrics as validity, uniqueness, novelty and diversity can easily be diverted through small modifications on an original SMILES string (Renz et al., 2019). Though such challenges remain common in other domains such as language and image, we encourage users to work closely with application-driven experts such as medicinal chemists or health institutes.

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

## A. Property Distributions for EGFR-MET and PIK3CA-mTOR

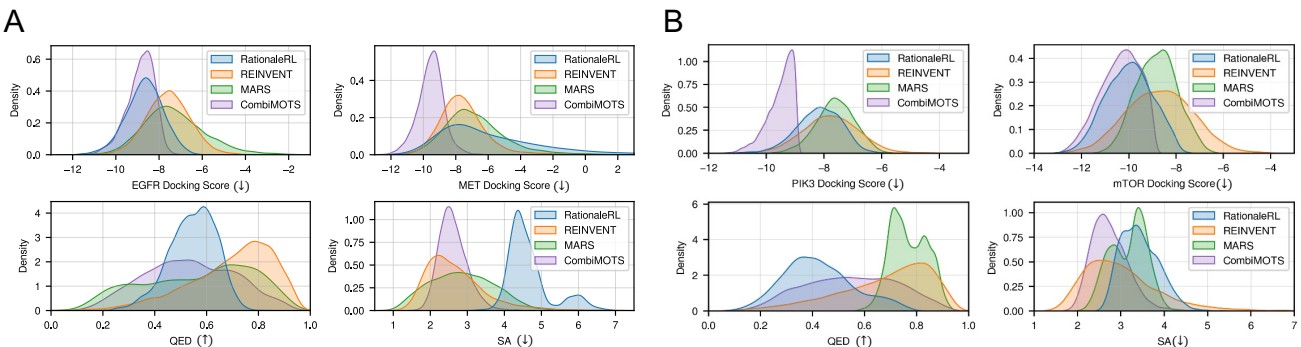

*Figure 7.* Normalized distributions of the generated molecules across dual-docking scores, QED and SA scores on the (A) EGFR-MET and (B) PIK3CA-mTOR target-pairs.

## B. Tables and Property Distributions - Ablation Studies

*Table 5.* Performance comparison of fragmentation methods on the GSK3$\beta$-JNK3 task for 10,000 sampled molecules on three independent runs.

| Method | Metrics | | | |
|---|---|---|---|---|
| | Valid. (%) | Uniq. (%) | Novel. (%) | Div. (%) |
| BRICS | **100**$_{\pm0.00}$ | **100**$_{\pm0.00}$ | 98.54$_{\pm0.09}$ | 88.31$_{\pm1.67}$ |
| MiCaM | **100**$_{\pm0.00}$ | **100**$_{\pm0.00}$ | 98.26$_{\pm0.81}$ | 87.99$_{\pm1.34}$ |
| FGIB (base) | **100**$_{\pm0.00}$ | **100**$_{\pm0.00}$ | **99.96**$_{\pm0.01}$ | **88.67**$_{\pm0.52}$ |

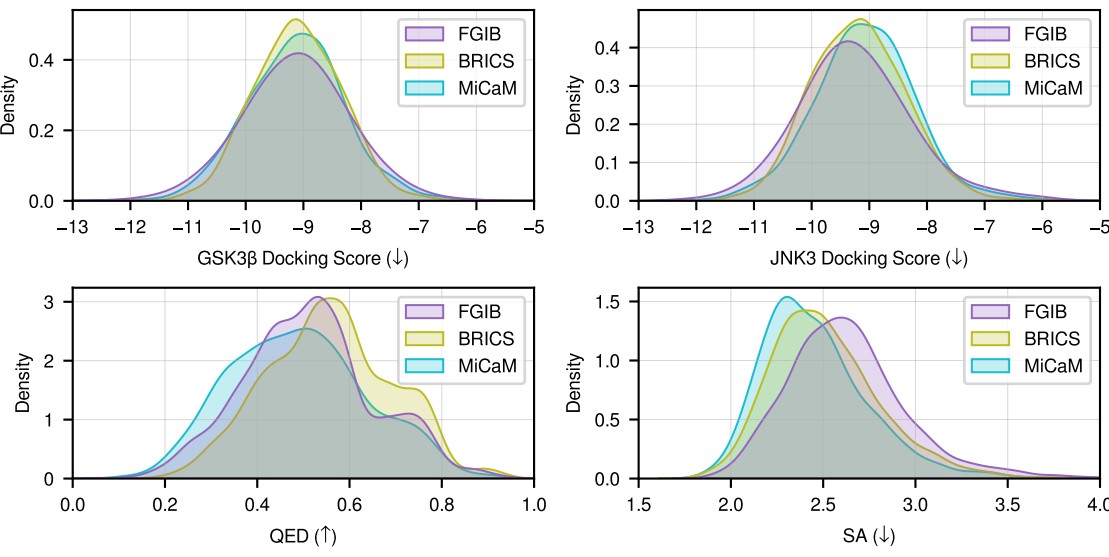

*Figure 8.* Distributions plots comparing fragmentation methods on the GSK3$\beta$-JNK3 task.

*Table 6.* Performance comparison across different threshold values based on 10,000 sampled molecules on the GSK3$\beta$-JNK3 task for three independent runs. Note the drop in Diversity for a threshold value of 0.6, reflecting redundancy related to a limited search space.

| Threshold | Metrics | | | |
|---|---|---|---|---|
| | Valid. (%) | Uniq. (%) | Novel. (%) | Div. (%) |
| 0.3 | $100_{\pm0.00}$ | $100_{\pm0.00}$ | $98.80_{\pm0.21}$ | $88.38_{\pm0.62}$ |
| 0.5 | $100_{\pm0.00}$ | $100_{\pm0.00}$ | $99.93_{\pm0.02}$ | $86.28_{\pm0.35}$ |
| 0.6 | $100_{\pm0.00}$ | $100_{\pm0.00}$ | $99.95_{\pm0.01}$ | $78.73_{\pm1.11}$ |
| 0.4 (base) | $100_{\pm0.00}$ | $100_{\pm0.00}$ | $\mathbf{99.96_{\pm0.01}}$ | $\mathbf{88.67_{\pm0.52}}$ |

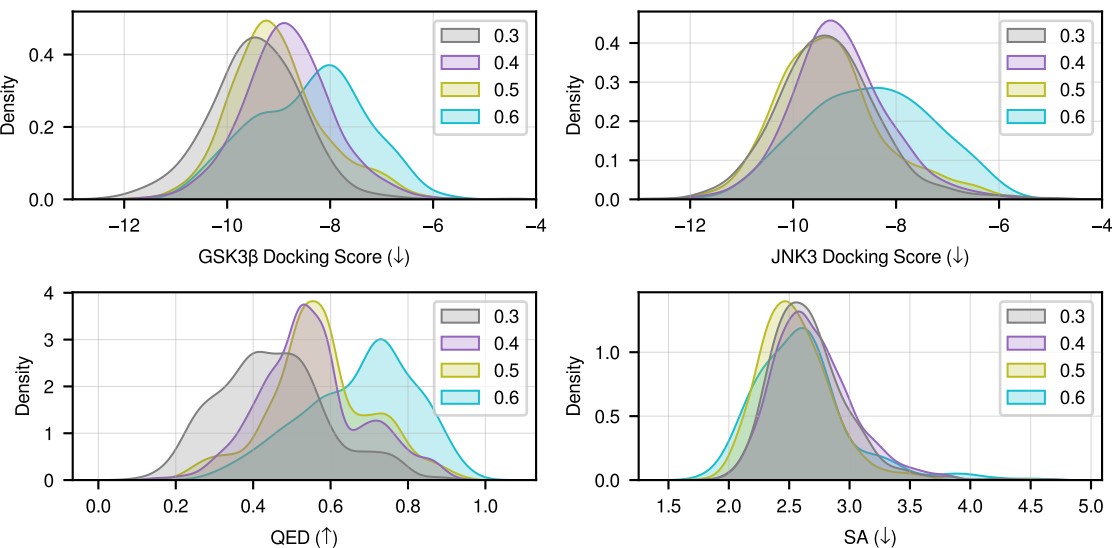

*Figure 9.* Distributions plots comparing Tanimoto similarity thresholds on the GSK3$\beta$-JNK3 task. A value of 0.3 converges slower in QED. Values of 0.4 and 0.5 perform similarly but 0.4 presents more dual-actives. From 0.6, the search space is too small to find better tradeoffs.

## C. t-SNE Visualizations

We provide t-SNE visualizations of the baseline models for all three tasks using ECFP4 fingerprints (2048 bits), a perplexity of 30, 1,000 iterations and the seed number 42. We plot Figures 10 to 12 using the same generation runs than for Figures 3 and 7.

Key observations are as follows:

- RationaleRL generations are clustered on all tasks. Its workflow revolves around (i) extracting and merging "rationales" from high-property compounds, then (ii) training & finetuning a graph completion module to obtain novel molecules from merged rationales. Generation occurs by auto-regressively complete the same structural cores, thus leading to poor diversity and similar fingerprints.

- REINVENT and MARS are not consistently generating sparse point clouds. For example REINVENT is performing poorly on the GSK3$\beta$-JNK3 task, and MARS on the EGFR-MET task where small clusters are forming.

- CombiMOTS generates sparse and wide point clouds across all tasks, indicating its consistency in generating novel and structurally diverse compounds.

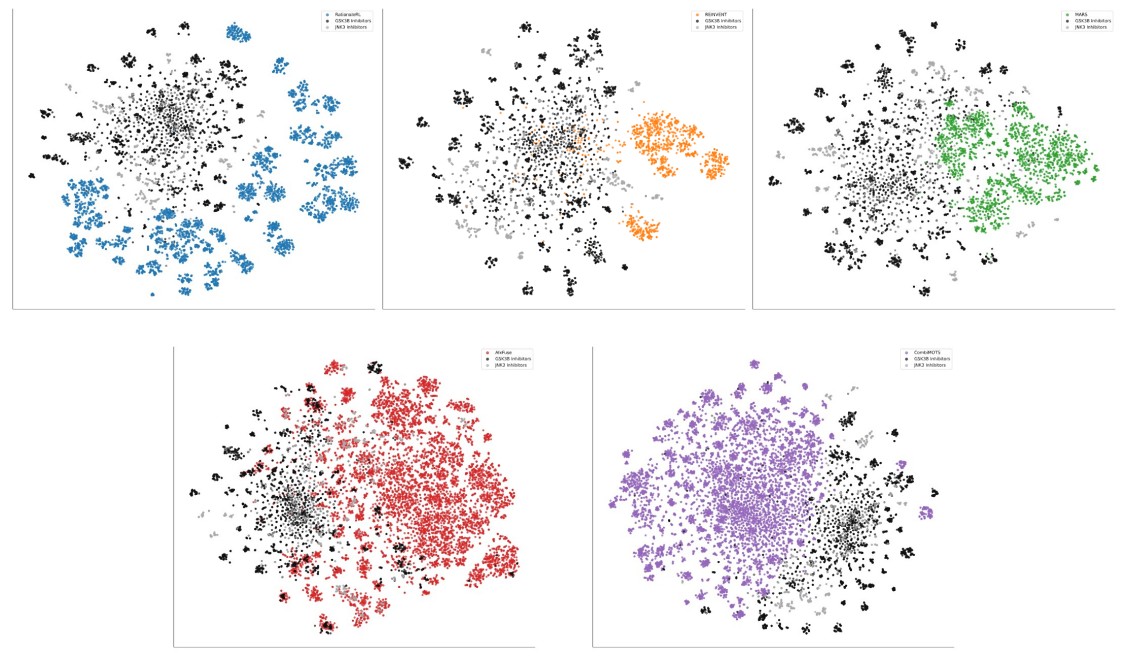

*Figure 10.* t-SNE distributions on the GSK3$\beta$-JNK3 task.

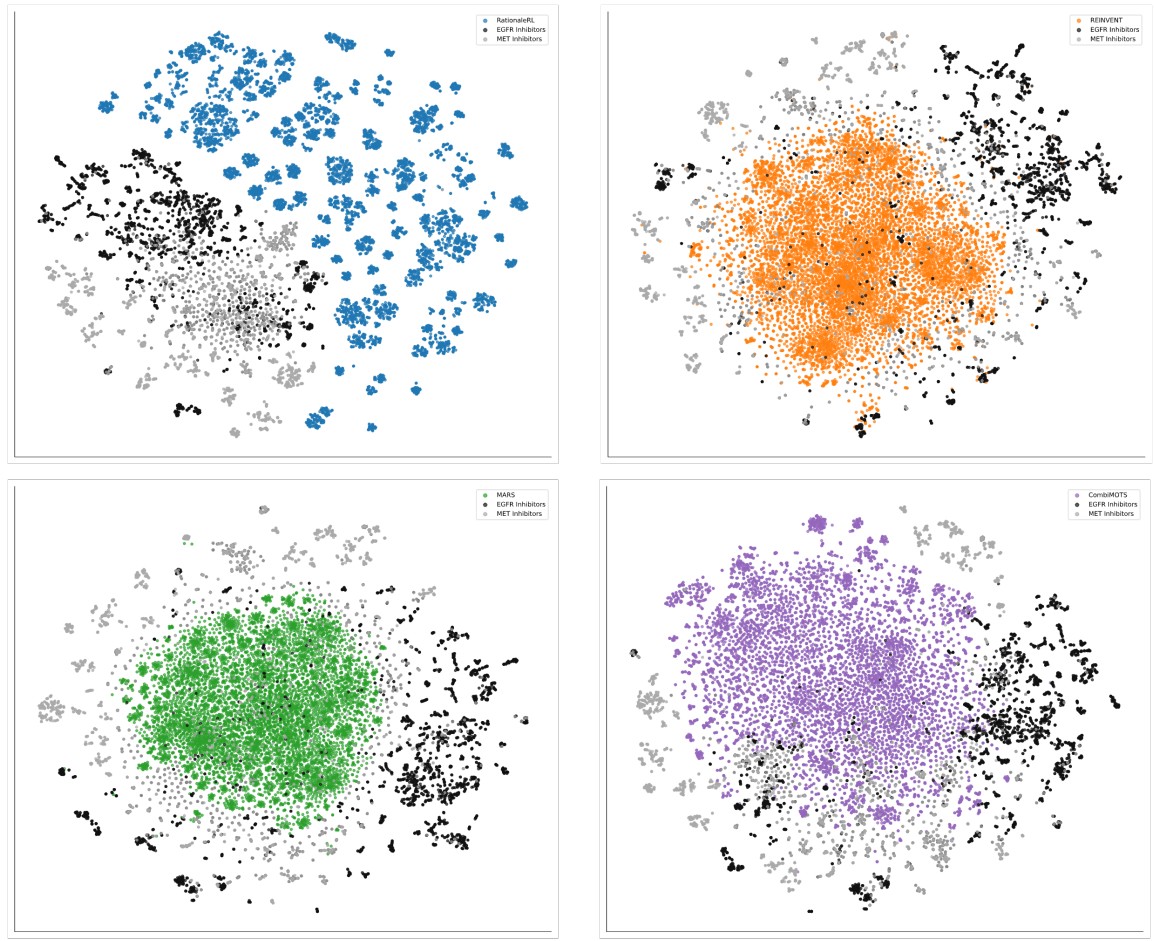

*Figure 11.* t-SNE distributions on the EGFR-MET task.

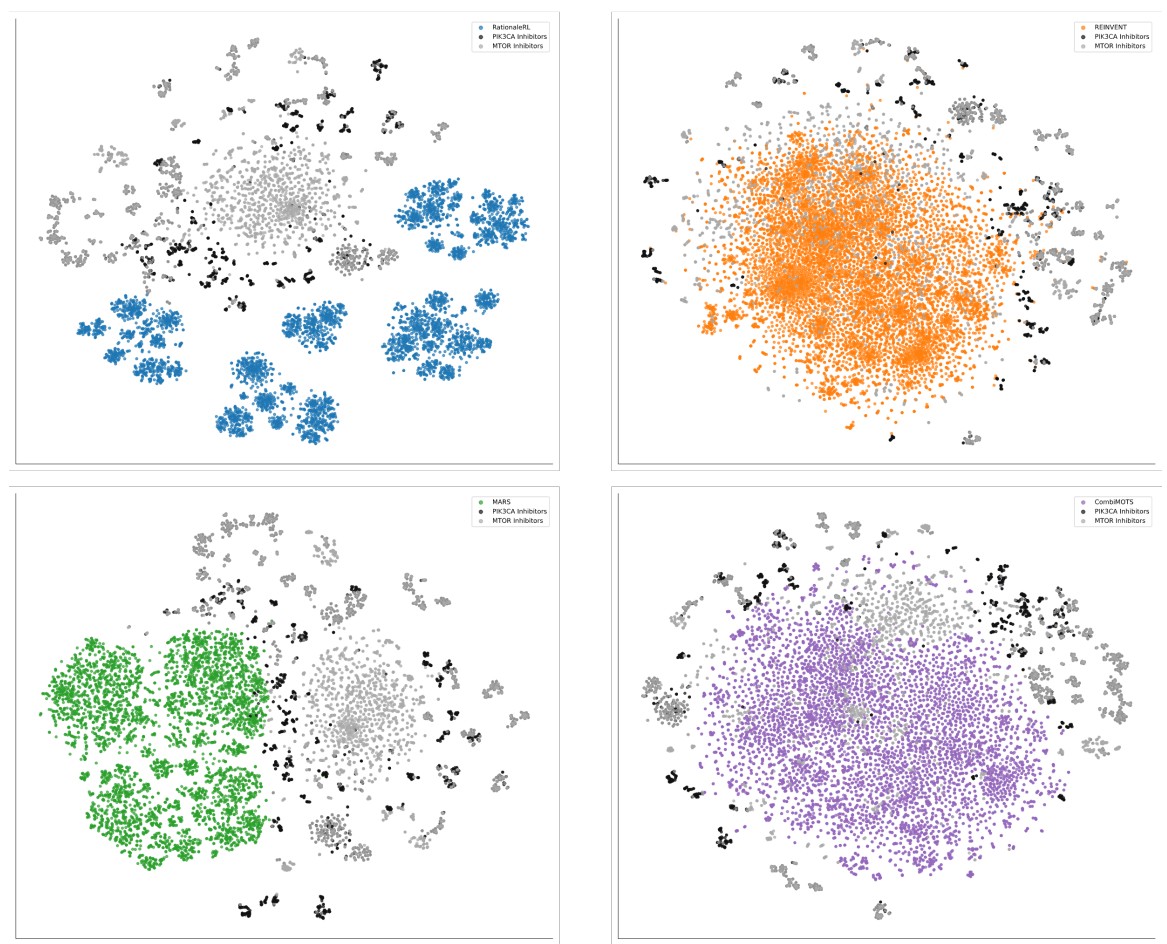

*Figure 12.* t-SNE distributions on the PIK3CA-mTOR task.

## D. Data Preparation

### D.1. Curation of EGFR-MET and PIK3CA-mTOR

We downloaded the ExCAPE-DB (Sun et al., 2017) from the following zenodo record and retained only entries with a Tax_ID of 9606 (Homo sapiens). Using RDKit, we then converted all molecular structures to canonical SMILES. Finally, we filtered out molecules that did not meet the following criteria—having fewer than 50 heavy atoms and containing only the elements H, B, C, N, O, F, P, S, Cl, Br, and I—and retained only those that satisfied these requirements (Li et al., 2018).

### D.2. Curation of CDK7 and Off-targets

We downloaded the PubChem database (as of 2024, November 25th) (Kim et al., 2016) to curate bioactivity data. We filtered out molecules having conflicting labels or which did not meet the following criteria—having a molecular weight (MW) below 1,000gr/mol, having more than 12 heavy atoms, not containing metal atoms or ions and whose activity was measured in at least two proteins—and retained only those that satisfied these requirements (Sun et al., 2017). The final curated data comprises a total of 18,478 molecules which statistics are specified in Table 7.

*Table 7.* Distribution of molecules based on their activity against CDK targets.

|          | CDK1  | CDK2   | CDK5   | **CDK7** | CDK9  | CDK12 | CDK13 |
|----------|-------|--------|--------|----------|-------|-------|-------|
| Active   | 1,358 | 2,249  | 988    | **923**  | 1,065 | 434   | 149   |
| Inactive | 733   | 15,592 | 15,805 | **530**  | 533   | 239   | 334   |
| Total    | 2,091 | 17,841 | 16,793 | **1,453**| 1,598 | 673   | 483   |

### D.3. Implementation Details

All experiments were done using an Intel Xeon Gold 6526Y and a single NVIDIA RTX A6000 GPU.

- **CombiMOTS Activity Predictors**: Following Swanson et al. (2024) we use Chemprop (D-MPNN).

- **Baseline Random Forest Classifiers**: For RationaleRL, REINVENT, MARS and MolSearch, we follow Li et al. (2018) to train random forest classifiers for each different target pair. The data splitting follows a random 80:20 (Train:Test) ratio, implemented with scikit-learn (Pedregosa et al., 2011) with a number of estimators of 100. RDKit is used to calculate the ECFP6.

- **Molecular Docking**: We use QuickVina-GPU-2.1 (Tang et al., 2024) as our oracle. Target structures and binding centers coordinates are curated through Protein Data Bank (PDB) and ligand preparation is made with OpenBabel (O'Boyle et al., 2011). We list the used PDB IDs for each target: GSK3$\beta$ (6Y9S), JNK3 (4WHZ), EGFR (1M17), MET (4MXC), PIK3CA (8V8I), mTOR (3FAP), DHODH (6QU7), ROR$\gamma$t (5NTP).

  For the CDK7 selective generation task (Appendix I.1), we used: CDK1 (4Y72), CDK2 (3PXF), CDK5 (7VDP), CDK7 (8PLZ), CDK9 (3TN8), CDK12 (7NXK), CDK13 (5EFQ).

- **Objective Scaling:** Contrary to the baselines, the implementation of vectorized properties implies (i) the non-necessity to scale the objectives, (ii) and the possibility to simultaneously maximize some objectives while minimizing others. For better readability in our code implementation, we still conform them to the standard [0,1] range to be maximized. Following Lee et al. (2023), we apply the following transformations on docking scores and synthetic accessibility scores:

$$\widehat{DS} \leftarrow \frac{-DS}{20} \text{ and } \widehat{SA} \leftarrow \frac{10 - SA}{9} \tag{Transf.}$$

- **Normalized R2-distance to the utopia point:** Following Kusanda et al. (2022), we consider $n$ objectives to be maximized. More specifically, our main setting uses four objectives to be maximized in range 0 to 1 (after Transf.), leading to a final range of 0 to 2. The metric is defined as:

$$D(y, y_{uto}) = \sqrt{\sum_{i=1}^{n} \left( \frac{max(0, y_{uto,i} - y_i)}{y_{uto,i}} \right)^2}. \tag{9}$$

# E. Additional Experiments

## E.1. Objective Settings of CombiMOTS

To investigate the performance of our objective design, we compare the base settings of CombiMOTS against two modified settings: a linearized version not using Pareto optimality (E.1.1) and a six-objective version additionally considering QED and SA scores (E.1.2). We also dwell into the prioritization of docking score over QED and SA in E.1.3.

### E.1.1. SCALARIZATION

We empirically showcase how Pareto principles (Luc, 2008) help finding optimal tradeoffs in a given search space. We implement a scalarized version of CombiMOTS using the same four objectives (predicted dual-activities and dual-docking scores) with equal weights set to *0.25* (no *a priori* assumption), and run $N_{rollout}$= 10,000 iterations on the GSK3$\beta$-JNK3 task, using the exact same search space as our base setting. Figure 13 compares the normalized distribution of the generated molecules.

A first observation is that after 10,000 iterations are completed, the scalarized MCTS only found 1,488 dual actives, compared to 3,662 using Pareto optimization.

A second observation is that the scalarized version finds molecules with better QED score, but worse in all other metrics. As expected, this evidence supports the fact even with aligned objectives and identical search space, the naive aggregation of multiple objectives cannot effectively find compounds with balanced properties, making these approaches unsuitable for complex multi-objective settings.

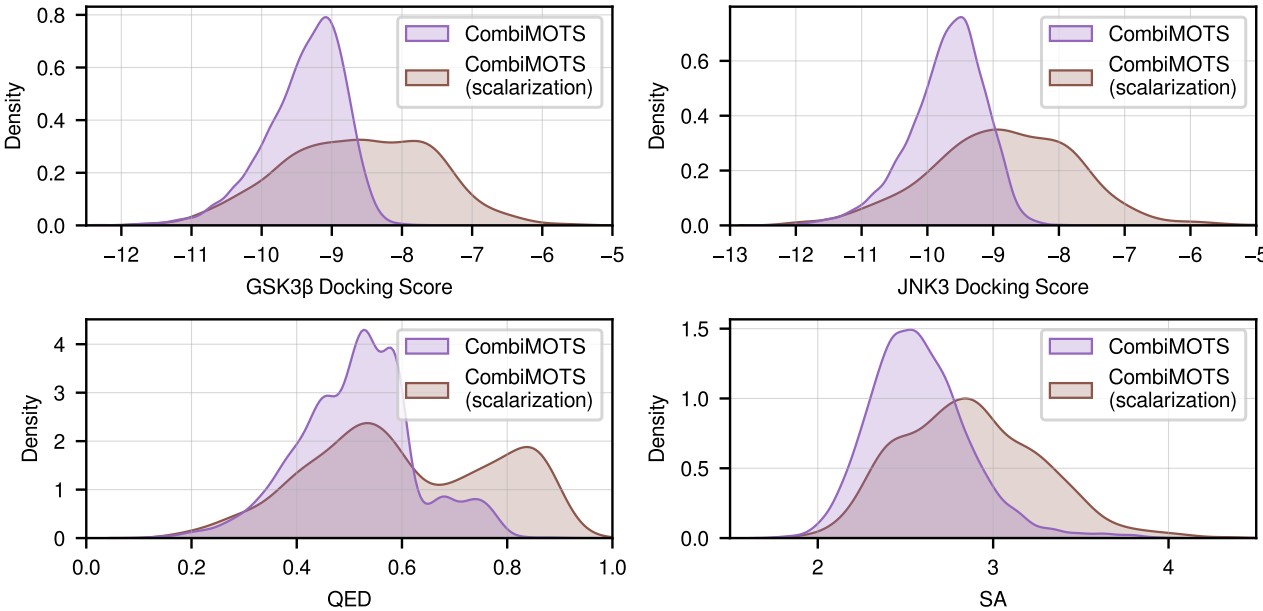

*Figure 13.* Normalized distributions of CombiMOTS against its scalarized version on the GSK3$\beta$-JNK3.

### E.1.2. SCALABILITY - NUMBER OF OBJECTIVES

We investigate the impact of considering more constraints in our framework. The theoretical analysis in Appendix L discusses convergence bounds and regret of the PUCB formula, as well as the effects of dimensionality (number of objectives). Intuitively, Theorem L.4 suggests that using more objectives leads to slower convergence due to larger, thus harder-to-distinguish local Pareto fronts. We aim to approach optimal solutions within a reasonable time scale, hence the need to properly select objectives, efficient oracles and search space size.

We implement a version of CombiMOTS using the same four objectives (predicted dual-activities and dual-docking scores) as our base setting, adding QED and SA scores as two additional objectives. We run $N_{rollout}$= 20,000 iterations on the GSK3$\beta$-JNK3 task, using the exact same search space as our base setting. Figure 14 compares the normalized distribution of the generated molecules.

A first observation is that after 20,000 iterations are completed, the six-objective CombiMOTS implementation found 5,748 dual actives, compared to 8,606 using only four objectives. Compared to the scalarized approached discussed in Appendix E.1, the number of predicted dual-actives is relatively closer to our setting, hinting at the better exploration of Pareto MCTS given multiple objectives.

A second observation is that the six objective version finds molecules with similar QED and SA score distribution as our base setting, but worse in docking scores. This behavior corroborates that adding objectives leads to more 'directions' to explore and induces slower convergence toward Pareto optimal solutions. Table 9 supports this claim.

We also note an additional computational overhead (from an average 11s/rollout to 25s/rollout), mainly due to the QED and SA oracles, slowing individual rollouts. Appendix F.2 further analyzes the complexity of our tree traversal.

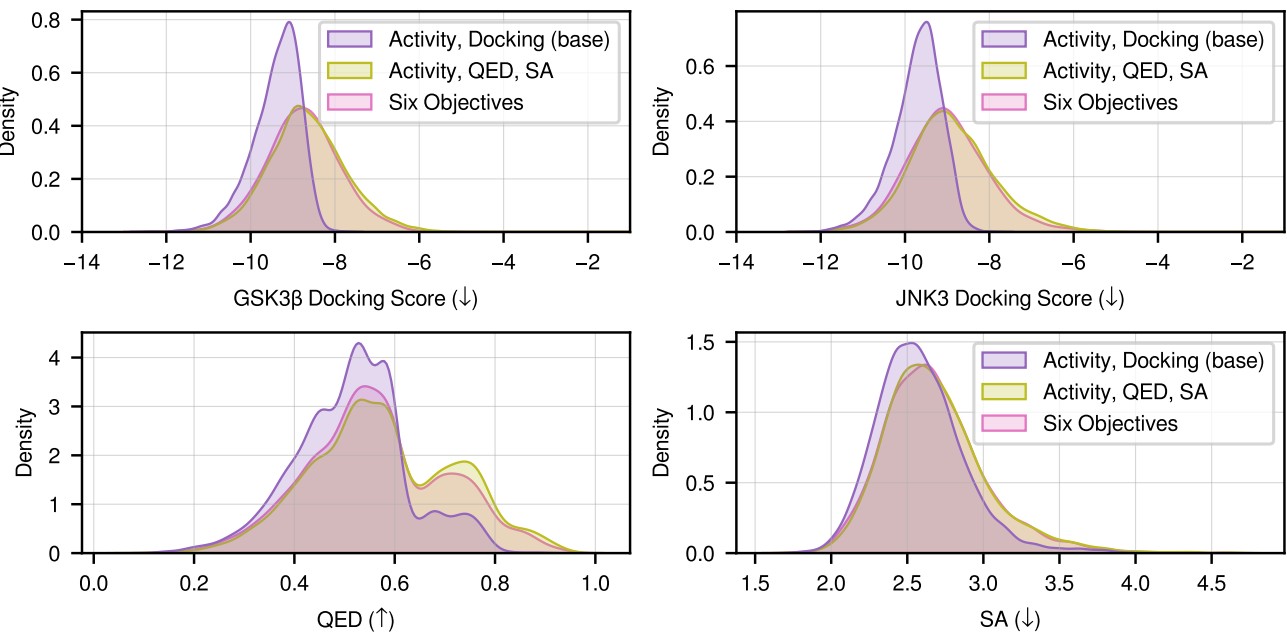

*Figure 14.* Normalized distributions of CombiMOTS against a version prioritizing QED/SA over docking score, and its six objectives version on the GSK3$\beta$-JNK3 task.

### E.1.3. SCALABILITY - IMPORTANCE OF DOCKING SCORE

The previous section (E.1.2) interrogates on the need to prioritize some objectives over others. Our base setting utilizes four objectives being predicted biological activity and docking score for both targets. We first emphasize how the structural and mechanistic relevance of docking lead to unveiling better candidate compounds, before empirically supporting this claim.

Docking score is critical in drug design as it directly evaluates ligand-target binding affinity, which is essential for therapeutic efficacy. Unlike QED/SA, which focus on general drug-likeness or synthetic feasibility, docking scores are structurally and energetically tied to the biological activity of the molecule. Studies have shown that incorporating docking scores during molecular generation leads to higher binding affinity and improved hit identification (Chenthamarakshan et al., 2020), whereas QED/SA cannot ensure target specificity (Agu et al., 2023; Xu et al., 2024). Therefore, docking score should be prioritized in early drug discovery stages, with QED and SA used later for optimization (Xue et al., 2025).

Table 8 and Figure 14 show how prioritizing QED/SA over docking scores compares to our base setting. We observe that not optimizing docking scores leads to a significant decrease of overall quality. Though QED is marginally improved, SA remains almost unaffected while binding affinity gets considerably worse: our base setting still leads to acceptable scores even without explicitly optimizing QED/SA.

On the GSK3$\beta$-JNK3 task, we show in Table 9 the size of the first six Pareto fronts from 10k randomly sampled products, as well as the number of Pareto fronts in three different objective settings: prioritizing docking score over QED/SA, prioritizing QED/SA over docking score and using all six objectives. We observe that as expected, using six objectives leads to fewer Pareto fronts which are individually more populated. Also, using docking docking score interestingly enables better differentiation between Pareto fronts, more numerous and individually less populated.

Though these results are empirical, they perfectly align with the necessity to consider both target affinity and molecular properties within the limitations of dimensionality scaling for Pareto MCTS.

*Table 8.* Performance of our base settings compared to using QED, SA as objectives on the GSK3$\beta$-JNK3 task for three independent runs.

| Setting | Metrics | | | |
|---|---|---|---|---|
| | Valid. (%) | Uniq. (%) | Novel. (%) | Div. (%) |
| QED, SA | **100**$_{\pm 0.00}$ | **100**$_{\pm 0.00}$ | 99.92$_{\pm 0.00}$ | **90.32**$_{\pm 0.95}$ |
| Docking Scores (base) | **100**$_{\pm 0.00}$ | **100**$_{\pm 0.00}$ | **99.96**$_{\pm 0.01}$ | 88.67$_{\pm 0.52}$ |

*Table 9.* Number of molecules across Pareto fronts for different objective settings on the GSK3$\beta$-JNK3 task.

| Pareto Rank | Activity + Docking | Activity + QED + SA | Six Objectives |
|---|---|---|---|
| Rank 1 | 60 | 69 | 729 |
| Rank 2 | 135 | 204 | 1,416 |
| Rank 3 | 180 | 257 | 1,809 |
| Rank 4 | 233 | 370 | 1,771 |
| Rank 5 | 263 | 456 | 1,456 |
| Rank 6 | 316 | 502 | 1,153 |
| Number of fronts | 40 | 29 | 12 |

### E.2. Aligned Baselines

We investigate the performance of MARS and REINVENT when using dual docking scores as objectives instead of QED and SA, with equal weights set to 0.25. For REINVENT, we generate 10,000 molecules using the integrated DockStream tool. For MARS, we implement QuickVinaGPU-2.1 as an online oracle. However, as MARS generates $N$ samples at every step of a run, the computational cost of molecular docking does not allow to generate 10,000 compounds. We therefore only run MARS to generate 500 molecules and analyze the samples from the converged step. Figure 15 compares the normalized distribution of the returned molecules.

We observe once again the property imbalance of both MARS and REINVENT, supposedly due to their scalarized objective design. MARS generates compounds exhibiting a better docking score, but lower QED score, while REINVENT contrarily generates compounds with worse docking score for both targets but high QED scores. On the other hand, CombiMOTS is the only model consistently identifying compounds with high property tradeoffs.

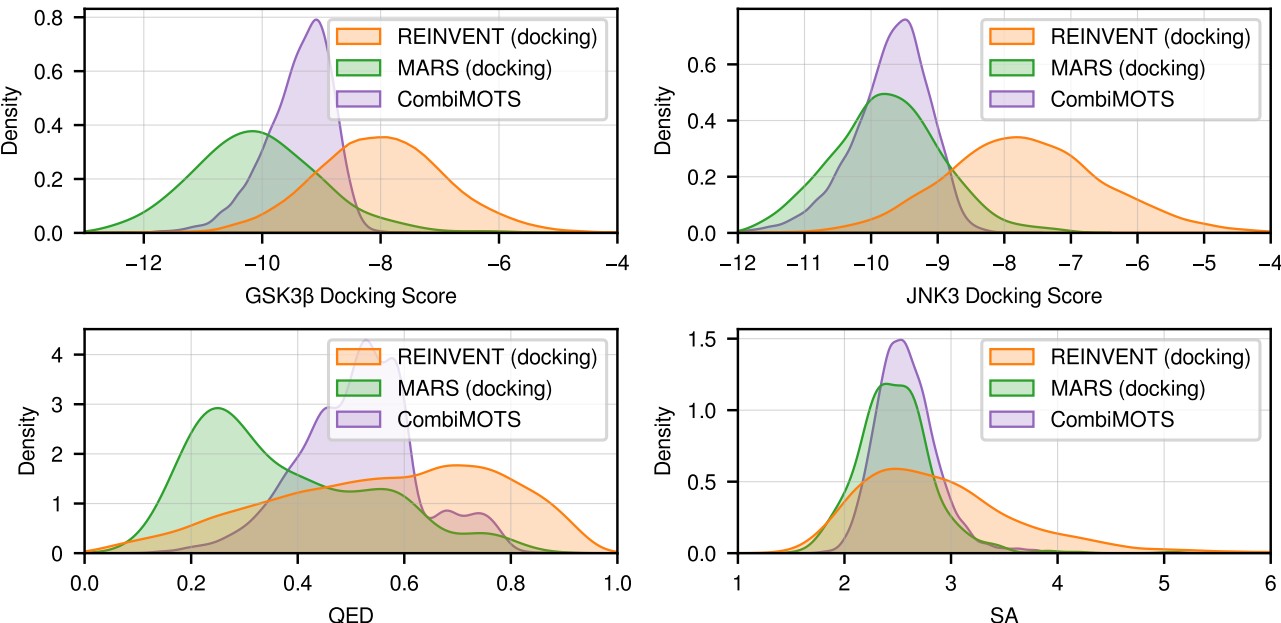

*Figure 15.* Normalized distributions of CombiMOTS against aligned versions of MARS and REINVENT on the GSK3$\beta$-JNK3 task.

### E.3. Impact of Data Availability

We investigate the performance of CombiMOTS against AIxFuse (Chen et al., 2024). AIxFuse is a method extracting pharmacophores using docking pose protein-ligand analysis, before collaboratively use RL and Active Learning (AL) for fragment assembly. It progressively learns how to fuse and dock the extracted fragments better. We run the notebooks following the original GitHub guidelines and generate 10,000 molecules with 4 iterations. Figure 3 compares performance on the GSK3$\beta$-JNK3 task, while Figure 16 compares performance on the DHODH-ROR$\gamma$t task proposed by the authors. Specifically, the DHODH-ROR$\gamma$t target pair was not curated by only using ground-truth experimental assay results. Due to data scarcity, Chen et al. (2024) make assumptions on the ChEMBL dataset to curate additional data.

On the GSK3$\beta$-JNK3 task (Figure 3), AIxFuse's performance is reasonably fair due to the high quality of the data. However, this does not hold true for the DHODH-ROR$\gamma$ target pair. In particular, docking scores of the generated molecules are particularly low. We note that in both tasks, AIxFuse exhibits a lower QED score than all baselines. This behavior could also be explained by the linear design of AIxFuse's objective function failing to find tradeoffs. In contrast, CombiMOTS still manages to find satisfying candidates, exhibiting once again the efficacy of our Pareto-based approach and complementary choice of objectives.

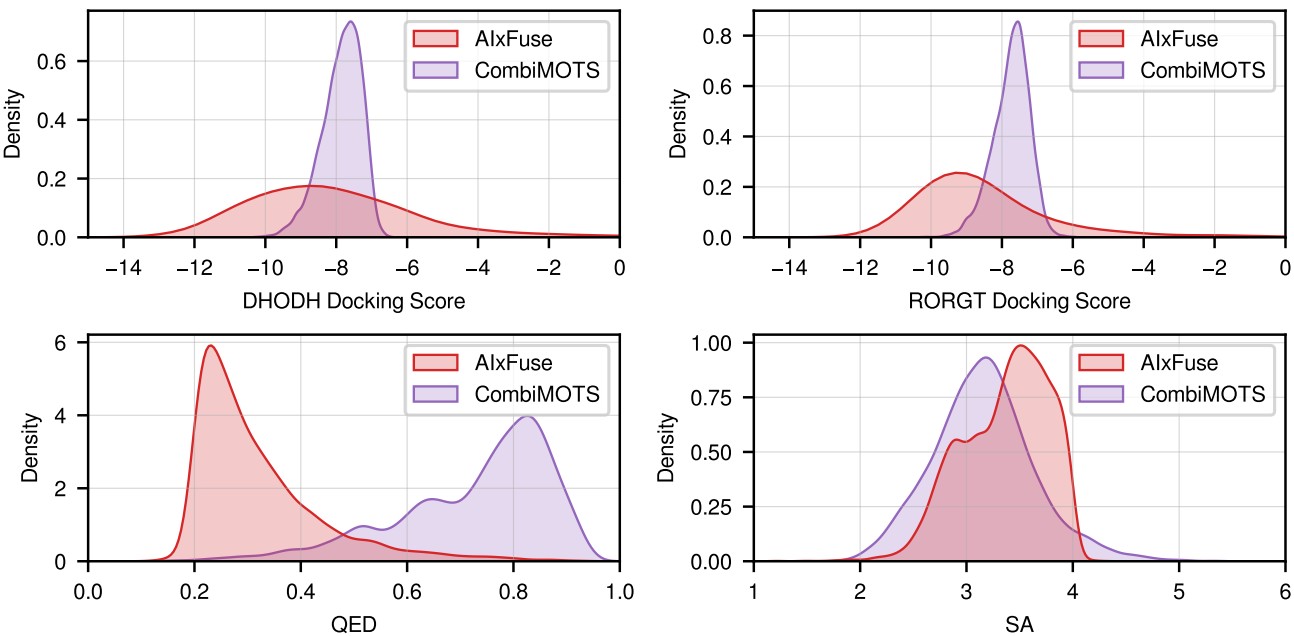

*Figure 16.* Normalized distributions of the generated molecules across dual-docking scores, QED and SA scores on the DHODH-RORγt target pair.

**E.4. Comparison against Pareto MCTS baselines**

We investigate the performance of CombiMOTS against other Pareto MCTS methods: Mothra (Suzuki et al., 2024) and MolSearch (Sun et al., 2022).

Mothra is a multiobjective molecular generation system that integrates recurrent neural networks (RNN) with Pareto MCTS to simultaneously optimize multiple properties like target protein affinity, drug-likeness, and toxicity. Compared to CombiMOTS, Mothra is natively limited to single-target RNN decoding. We adapt Mothra to support dual-target settings by running molecular docking for both proteins, and using the returned scores during generation.

MolSearch is a Pareto MCTS-based framework that starts with existing molecules and uses a two-stage search strategy with transformation rules derived from compound libraries to gradually modify them for multi-objective optimization. Compared to CombiMOTS, MolSearch is natively focused on dual-target optimization using RF classifiers, rather than de novo generation. We adapt MolSearch to the EGFR-MET and PIK3CA-mTOR tasks by training RF classifiers as detailed in Appendix D.3.

For both methods, we perform three independent runs following the original GitHub guidelines show in Table 10 and Figure 17 metrics for generated molecules.

Mothra returns extremely few valid molecules, among which even less are dual actives: for the run displayed in Figure 17, 1,125/131,688 molecules are valid and 6 are dual actives. MolSearch returns sound candidates in terms of molecular properties but is has a high computational cost: the run of Figure 17 was interrupted after 57 hours and generated 493 molecules. For both methods, generated compounds exhibit weak binding affinity towards GSK3$\beta$ and JNK3, with Mothra performing poorly in SA score. Appendix F discusses how these compare in terms of computational cost.

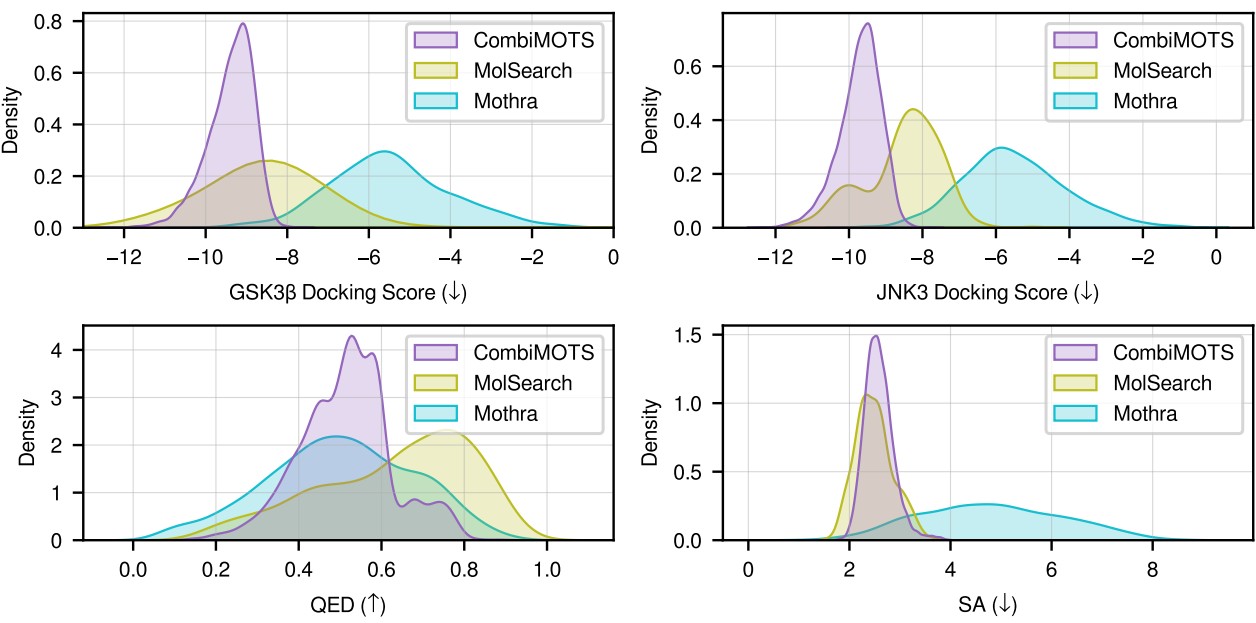

*Figure 17.* Normalized distributions of CombiMOTS against Mothra and MolSearch on the GSK3$\beta$-JNK3 task.

*Table 10.* Performance against additional Pareto MCTS baselines across target pairs. Results are obtained from three independent runs.

| Target Pairs | Models | Metrics | | | | |
|---|---|---|---|---|---|---|
| | | Valid. (%) | Uniq. (%) | Novel. (%) | Div. (%) | Act. SR (%) |
| GSK3$\beta$-JNK3 | MolSearch | $100_{\pm0.00}$ | $72.14_{\pm5.24}$ | $74.57_{\pm15.75}$ | $75.99_{\pm1.24}$ | $\mathbf{98.34}_{\pm1.20}$ |
| | Mothra | $0.68_{\pm0.15}$ | $99.82_{\pm0.03}$ | $\mathbf{100}_{\pm0.00}$ | $\mathbf{95.04}_{\pm0.25}$ | $0.52_{\pm0.18}$ |
| | CombiMOTS | $\mathbf{100}_{\pm0.00}$ | $\mathbf{100}_{\pm0.00}$ | $99.96_{\pm0.01}$ | $88.67_{\pm0.52}$ | - |
| EGFR-MET | MolSearch | $100_{\pm0.00}$ | $81.64_{\pm0.46}$ | $75.83_{\pm7.17}$ | $82.48_{\pm2.13}$ | $\mathbf{92.56}_{\pm3.84}$ |
| | Mothra | $0.92_{\pm0.45}$ | $99.70_{\pm0.11}$ | $\mathbf{99.99}_{\pm0.02}$ | $\mathbf{94.74}_{\pm0.58}$ | $52.92_{\pm8.08}$ |
| | CombiMOTS | $\mathbf{100}_{\pm0.00}$ | $\mathbf{100}_{\pm0.00}$ | $99.81_{\pm1.62}$ | $90.75_{\pm0.40}$ | - |
| PIK3CA-mTOR | MolSearch | $100_{\pm0.00}$ | $80.62_{\pm2.76}$ | $\mathbf{100}_{\pm0.00}$ | $83.76_{\pm2.16}$ | $\mathbf{89.22}_{\pm0.87}$ |
| | Mothra | $0.69_{\pm0.34}$ | $99.72_{\pm0.01}$ | $\mathbf{100}_{\pm0.00}$ | $\mathbf{95.35}_{\pm1.06}$ | $85.32_{\pm1.82}$ |
| | CombiMOTS | $\mathbf{100}_{\pm0.00}$ | $\mathbf{100}_{\pm0.00}$ | $99.99_{\pm0.01}$ | $90.88_{\pm0.36}$ | - |

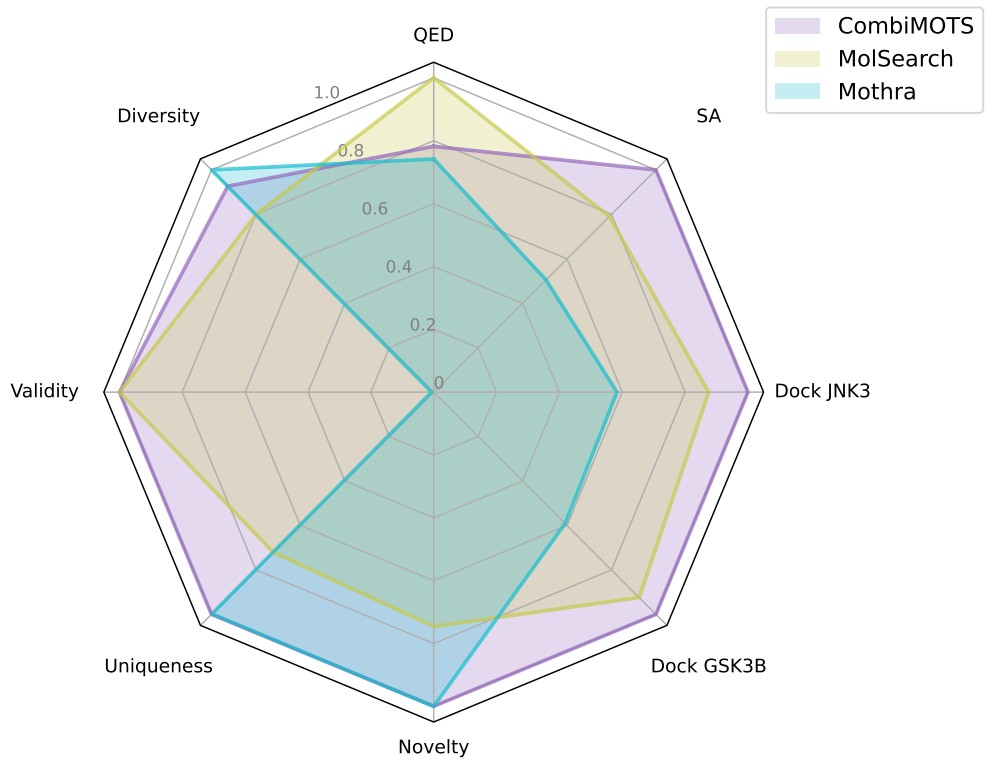

*Figure 18.* Radar chart on the GSK3$\beta$-JNK3 task against Mothra and MolSearch. We report average values of originality metrics and median values of quality metrics, normalized to the best score on each axis.

# F. Runtime and Complexity Analysis

In this section, we first report how the empirical runtime of all baseline methods compare against CombiMOTS in F.1, then provide in F.2 a complexity analysis of the Pareto MCTS algorithm implemented in CombiMOTS.

## F.1. Runtime Comparison

Note that RationaleRL finetuning step takes over 2 hours per epoch if the number of rationales is large. Among PMCTS methods (in gray), CombiMOTS is significantly more efficient, and just requires to train oracles if needed (below 10 minutes). Notably for Mothra, the original work performs 14-days experiments using two Intel Xeon E5-2680 V4 processors and four NVIDIA Tesla P100 GPUs, which represents far more resource than what CombiMOTS requires. For MolSearch, the original work reports an average of 0.4-1.0 hours per molecule in both HIT-MCTS and LEAD-MCTS stages using TITAN RTX GPUs (24GB), which is intractable when exploring large chemical spaces.

*Table 11.* Approximate runtimes on the GSK3$\beta$-JNK3 task.

| Model | Pretraining | Finetuning | Sampling/Searching Time (per iteration/rollout) | Total Search Time | Total Generations |
|---|---|---|---|---|---|
| RationaleRL | Yes | Yes | $\sim$10s/200 samples | $\sim$1min | 10k |
| MARS | Yes | No | $\sim$8min/step of 10,000 samples | $\sim$6hrs | 10k |
| REINVENT | Yes | Yes | below 1s/sample | $\sim$1min | 10k |
| Mothra | Yes | Yes | $\sim$17min/iteration | $\sim$30hrs (interrupted) | 131k, below 1% valid |
| MolSearch | Yes | No | $\sim$23min/iteration | $\sim$57hrs (interrupted) | 493 |
| CombiMOTS | (Oracles) | No | $\sim$9s/iteration | $\sim$4days, 6hrs | 79k |

## F.2. Complexity Analysis

As Appendix F.1 presents approximate runtimes, we further investigate the algorithmic complexity of CombiMOTS during the tree search. We consider our problem setting, which details are explicitely described in Appendix L.1.

We remind the following notations: $n_{children}$ the number of children from a parent node $node$, $B$ the set of building blocks, $R$ the set of reactions, $Oracles(.)$ the ensemble of property predictors.

The initialization step runs in $\mathcal{O}(1)$. Then, during a single rollout:

- If the selected node is a product ($\mathcal{O}(1)$):
    - Returning the product property vector runs in $\Theta(Oracles(product))$ as it is never precomputed.

- Else:
    - The expansion node creates $n_{children}$ in $\mathcal{O}((node.molecules + Card(B)) * Card(R))$;
    - We compute node properties upon creation:
        * If the scores are precomputed (using a lookup hash table), assign them to $child\_node$ in $\mathcal{O}(1)$;
        * Else, compute them in $\mathcal{O}(Oracles(child\_node))$;
    - The Pareto selection step computes the ParetoUCB vector of all children nodes, running in $n_{children}$;
    - The backpropagation step runs in $\mathcal{O}(1)$ as the maximum depth of the synthesis in bounded by the maximum allowed synthetic steps in $R$ (using Enamine REAL Space, one to two steps).

Therefore, a single tree traversal is bounded by the slowest operation between $Oracles(product)$ and $n_{children} * Oracles(child\_node)$. Moreover, if scores of building blocks are precomputed, this bound is further reduced to the slowest operation between $Oracles(product)$ and $O(n_{children})$.

## G. Identified Interactions of Case Study 6.1

**Compound 1** Hydrophobic interactions with *VAL70, ALA83, LEU132, GLN185* (GSK3$\beta$) and *ILE70, LEU144, LEU148, MET149* (JNK3). Hydrogen bonds with *ASN64, VAL135* (GSK3$\beta$) and *LYS93, GLN155, LEU206* (JNK3)

**Compound 2** Hydrogen bonds with *ASP133, VAL135, ASN186* (GSK3$\beta$) and *MET149, GLN155* (JNK3). Hydrophobic contacts involving *ILE62, VAL70, TYR134* (GSK3$\beta$) and *ILE70, VAL78* (JNK3).

## H. Selected compounds of Case Study 6.2

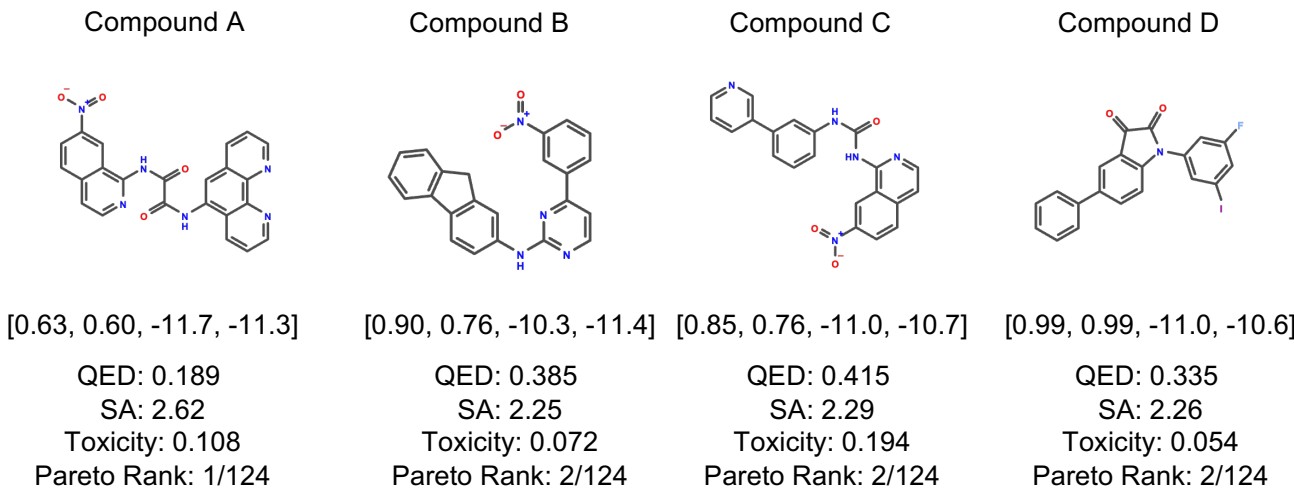

| Compound A | Compound B | Compound C | Compound D |
|---|---|---|---|
| [0.63, 0.60, -11.7, -11.3] | [0.90, 0.76, -10.3, -11.4] | [0.85, 0.76, -11.0, -10.7] | [0.99, 0.99, -11.0, -10.6] |
| QED: 0.189 | QED: 0.385 | QED: 0.415 | QED: 0.335 |
| SA: 2.62 | SA: 2.25 | SA: 2.29 | SA: 2.26 |
| Toxicity: 0.108 | Toxicity: 0.072 | Toxicity: 0.194 | Toxicity: 0.054 |
| Pareto Rank: 1/124 | Pareto Rank: 2/124 | Pareto Rank: 2/124 | Pareto Rank: 2/124 |

*Figure 19.* The four selected compounds for toxicity prediction. The vector values account for GSK3$\beta$-JNK3 predicted activities, and GSK3$\beta$-JNK3 predicted docking scores, respectively.

## I. Generalizability - Broader Applications

We discuss in this section how CombiMOTS could be adapted to other multiobjective tasks. A key advantage of using Pareto-based tree structure and nodes is to flexibly design property vectors to be either maximized or minimized. To exemplify this applicability, we conduct a preliminary case study for Cyclin-dependent kinase 7 (CDK7) selectivity in Appendix I.1, an ablation study of toxicity optimization (extending the observations made in Section 6.2) in Appendix I.2 and briefly debate protein-protein interaction modulator generation in Appendix I.3.

### I.1. Selective Molecular Generation

Selective molecular generation aims to identify compounds that are active only towards a given target. The intricacy of this task lies in avoiding off-targets often related to the main target protein, limiting the effectiveness of drugs or even risking adverse effects.

Prior works (Fisher, 2005; Sava et al., 2020) established CDK7 as a regulator of both cell cycle and global transcription, supporting its inhibition as a potential way to target deregulated cell proliferation. This motivated further research to identify off-targets, ultimately attempting to find selective inhibitors more relevant to polypharmacology for complex diseases (Olson et al., 2019; Constantin et al., 2023). We follow these studies to identify candidates selectively inhibit CDK7 while being inactive to six off-target kinases: CDK1, CDK2, CDK5, CDK9, CDK12, CDK13. Appendix D.2 contains details on the data curation.

In parallel, efforts were also made in targeted transcription regulation in cancer. Notably, Kwiatkowski et al. (2014) successfully discovered and characterized a covalent CDK7 inhibitor which achieved greater selectivity through its ability to target a cysteine residue outside of the canonical kinase domain.

Thanks to these findings, we adapt CombiMOTS through two initiatives:

- During the tree traversal, selectivity can be translated as a maximization objective towards CDK7 activity while minimizing all off-targets' activity. We train random forest predictors for each kinase using the curated data in a 8:1:1 training/validation/test ratio.

- The search space reduction step can be applied to preliminarily select building blocks susceptible to react into cystein-targeting products. Inspired by Huang et al. (2022), we identify four substructures from known inhibitors leading to 81 similar initial building blocks corresponding to over 2.3M possible products.

We propose the following pipeline:

1. From the four initial substructures, identify similar building blocks;

2. Run a 200k rollout tree search, only guided by CDK7 (maximize), CDK2 and CDK9 (minimize) bioactivity predictors. Note that we only select three objectives to converge faster towards Pareto optimal solutions;

3. We (re)-perform *post-hoc* bioactivity predictions for all kinases, and only retain molecules with predicted values above $0.5$ for CDK7 and below $0.5$ for off-targets;

4. Optionally perform *post-hoc* docking simulation for added practical information;

5. Apply industrial and medicinal filters to obtain a final list of candidates. We retain molecules with the following criteria: $-0.4 \leq LogP \leq 5.6$ (Ghose et al., 1999), $250 \leq MW \leq 500$, less than 5 Hydrogen Bond Donors (HBD), less than 10 Hydrogen Bond Acceptors (HBA) (Lipinski's Rule of Five), less than 10 rotatable bonds, $50 \leq TPSA \leq 140$Å (Veber's rule) and does not contain imine group (Kalgutkar, 2019) chiral center or stereocenter.

Table 12 summarizes the key described steps and Figure 20 shows examples of final candidates.

*Table 12.* Schematic workflow of our case study on CDK7 specific inhibitor search.

| Step | Statistic |
|---|---|
| Initialization | (2 known inhibitor scaffolds, 2 warheads) |
| Search Space Reduction | 81 building blocks, 2.3M possible products |
| Tree Search (200k rollouts) | 348k generations |
| Filter - Selective Activity | 3,491 candidates |
| *Post-hoc* docking simulation | - |
| Filter - (from medicinal chemistry) | 1,554 final candidates |

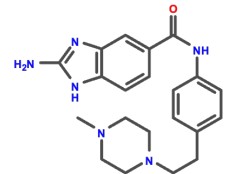

[0.7, 0.49, 0.48, 0.24, 0.42, 0.36, 0.16]     [0.68, 0.49, 0.36, 0.18, 0.42, 0.37, 0.20]     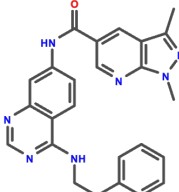 [0.66, 0.42, 0.47, 0.24, 0.44, 0.20, 0.11]

*Figure 20.* Example of found molecules active to CDK7 and inactive to off-targets. Vectors are predicted activities towards CDK7, CDK1, CDK2, CDK5, CDK9, CDK12, CDK13 respectively.

## I.2. Toxicity Optimization

We run 50k rollouts on the GSK3$\beta$-JNK3 task with an added objective being the predicted toxicity (to be minimized). We use the same Chemprop predictor trained in Section 6.2 during the tree search. As expected, we observe in Figure 21 that overall generations are less toxic and present a notable increase in density of molecules with very low toxicity (below 0.2).

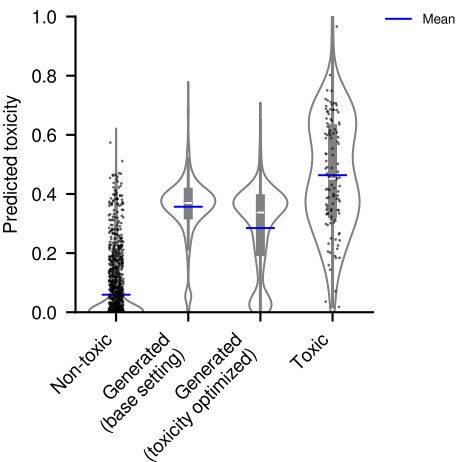

*Figure 21.* Chemprop predictions of molecules generated while aiming to minimize toxicity.

## I.3. Protein-Protein Interaction Modulator Generation

In this paper, we mainly tackle the task of identifying dual inhibitors by targeting two proteins related to the same disease. An other field of application differing from traditional protein-ligand binding, would be the area of protein-protein interaction. Though both interactions rely on similar physical forces, PPIs exhibit distinct physicochemical properties, such as generally larger interface areas or more complex geometry. In particular, developing small molecules tailored to PPI interfaces remains a challenge. Few recent works (Wang et al., 2024; Sun et al., 2024) implement interface-aware generative frameworks by considering hot-spot residues or known PPI complexes, enriching the landscape of SBDD.

We could envision CombiMOTS to be adapted to this task by either setting appropriate biological targets, or by aiming to mimic pharmacophoric features of hot-spot residues through the use of structure-oriented scorers during the tree traversal. We leave such investigations to future works.

## J. Retrosynthesis Study

We quantitatively assess the utility of using industrial databases like Enamine REAL as the initial search space of Combi-MOTS by discussing the pitfalls of synthesizability metrics in drug design. As generative models construct or optimize molecular structures, the general purpose of such metrics is to quantify the feasability of experimentally synthesizing the proposed candidates. Synthesizability metrics can roughly be distributed into three categories:

- **Rule-based** metrics (e.g., SAscore) make predictions out of deterministic features (motifs, molecular mass, etc.). Though they are intuitive and computationally efficient methods, they often fail to capture real-world constraints.

- **Retrosynthetic-based** approaches are more interpretable as they recursively break down a target compound into industry-available precursors. However, they often lead to consequent computational overheads and are limited to pre-defined reference stocks. For example, Aizynthfinder (Genheden et al., 2020) is limited to the ZINC database. Our approach is based upon the opposite idea: from a set of known precursors, we attempt to find Pareto optimal reaction products. This enables us to flexibly explore any chemical space at disposal, instead of iteratively testing known synthetic routes for a given target.

- **Learning-based** methods (e.g., RAscore (Thakkar et al., 2021), BR-SAscore (Chen & Jung, 2024)) have the advantage of being context-adaptable but become inherently dependent on the quality, format and availability of the training data.

We first propose to highlight the inconsistencies across these metrics by evaluating the generations from CombiMOTS, RationaleRL, MARS and REINVENT, as well as Enamine building blocks and COMPAS-3 (Wahab & Gershoni-Poranne, 2024) as representatives of easy-to-synthesize and hard-to-synthesize molecules, respectively. Notably, the COMPAS-3 dataset exclusively contains (poly)-cyclic compounds. We plot and report the molecular distributions in Figure 22.

As expected, we observe several inconsistencies: COMPAS-3 is deemed synthesizable by SA and RAscore but not by BR-SAscore. Enamine excels on RAscore but performs similarly or worse on other metrics. MARS unrealistically outperforms Enamine on SA/BR-SAscore. CombiMOTS outperforms baselines on RAscore but aligns with/worsens on others.

Additionally, we attempt to find Aizynthfinder routes for the final candidates found in the preliminary case study for CDK7 selective inhibitors (Appendix I.1). Because CombiMOTS leverages Enamine through their in-house validated synthetic procedures, we would normally expect all candidates to find be solvable, i.e. find a route for each compound. However, among 1,544 molecules, 162 (10.42%) are not solved by Aizynthfinder—due to its ZINC dependency. Figures 23 and 24 show examples of failure cases, as well as the route proposed by CombiMOTS.

These inconsistencies motivate the use of Enamine REAL Space over mere metric evaluation: though its true synthesizability still cannot be perfectly predicted, its empirical high synthesis success rate is more aligned with real-world feasability.

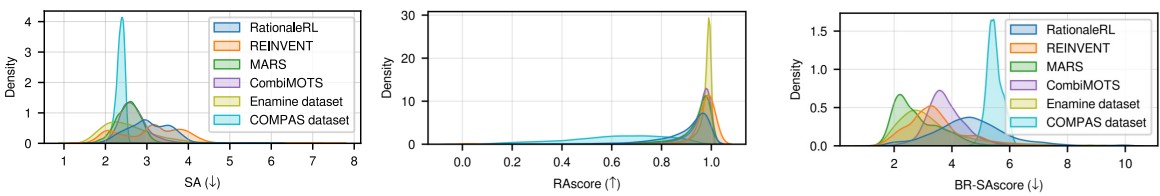

*Figure 22.* Plot distributions across synthesizability metrics. For generative models, we evaluate candidates from the GSK3$\beta$-JNK3 task.

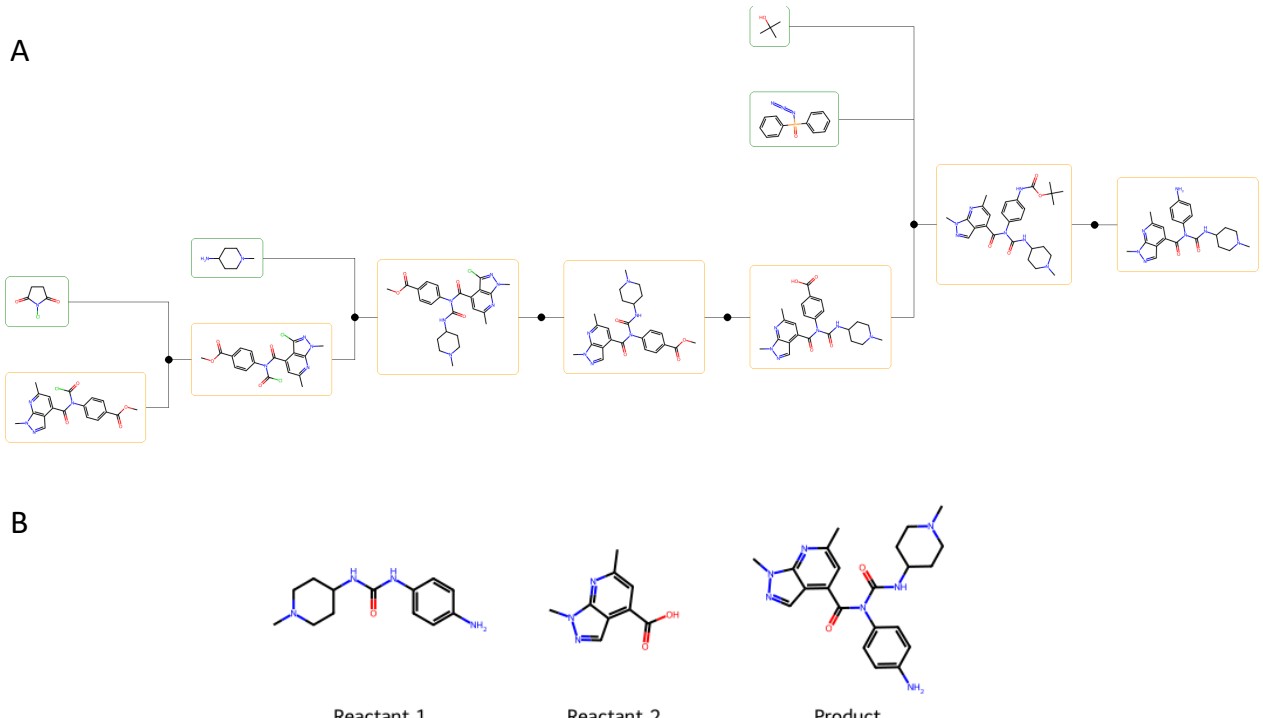

*Figure 23.* A) Unsolved routed of Aizynthfinder and B) Enamine building blocks found by CombiMOTS leading to the product. (Case 1)

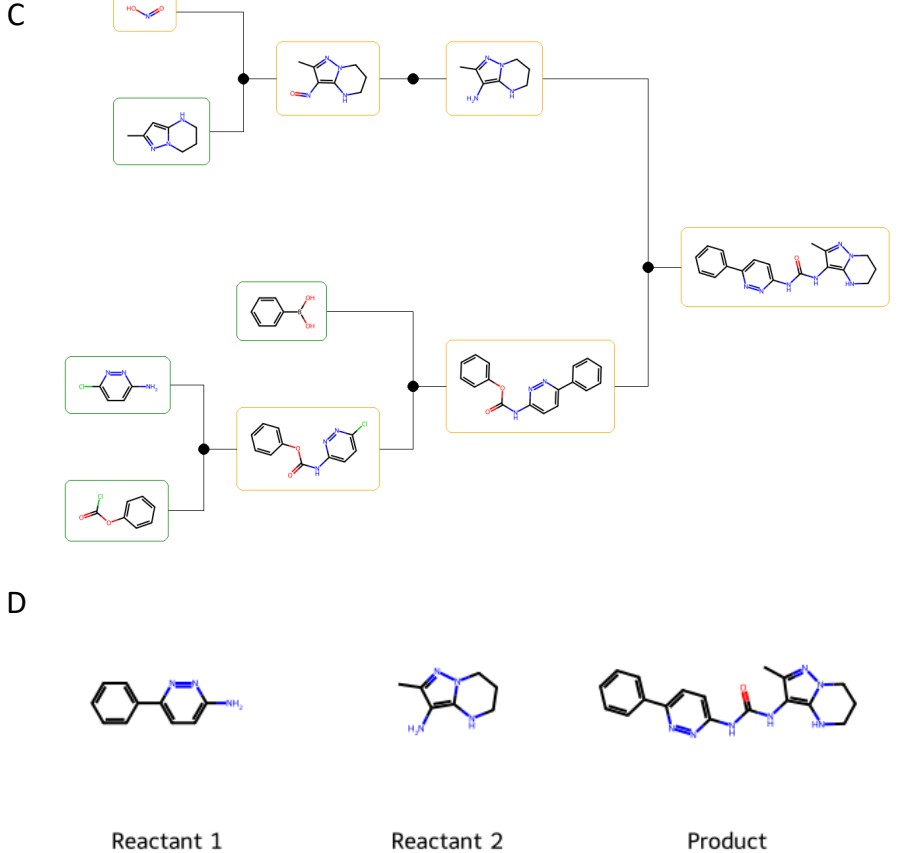

*Figure 24.* C) Unsolved routed of Aizynthfinder and D) Enamine building blocks found by CombiMOTS leading to the product. (Case 2)

## K. Detailed Implementation of Pareto MCTS in CombiMOTS

---

**Algorithm 2** Complete Pareto MCTS for CombiMOTS

---

**Input:** Synthesis tree $T$, Number of rollouts $n_{rollout}$
**Data:** Property predictors $Oracles$, building blocks $B$, reactions $R$, exploration weight $C$

    // Main Monte Carlo Tree Search Function
1  **Function** *ParetoMCTS($n_{rollout}$)*:
2      Initialize the tree $T.root$
3      **for** $i \leftarrow 1$ **to** $n_{rollout}$ **do**
4         *Rollout* ($T.root$)
5      **return** All visited nodes in $T$ that are reaction products

    // Rollout Function
6  **Function** *Rollout(node)*:
7      **if** *node is a reaction product* **then**
8         **return** $Oracles(node)$ // Terminal node = Backpropagation starts

9

10      *child_nodes* $\leftarrow$ *Expansion(node)* // Expand leaf node
11      *parent_visits* $\leftarrow \sum$(visits of all children)
12      *next_node* $\leftarrow$ *ParetoSelection(child_nodes, parent_visits)* // Select from the local Pareto front
13      $\vec{v} \leftarrow$ *Rollout(next_node)* // Recursive unroll = Simulation
14      **if** *next_node is a reaction product* **then**
15         $\vec{v} \leftarrow$ *element-wise-max($\vec{v}, Oracles(next\_node)$)*
16      Update *next_node* statistics (exploit score, visit count)
17      **return** Reward vector $\vec{v}$ // Backpropagation

18

    // Expansion Function
19  **Function** *Expansion(node)*:
20      Initialize $E \leftarrow \emptyset$
21      **foreach** *reaction* $\in R$ **do**
22         **if** *node.molecules compatible with reaction* **then**
23             Add one node per reaction product to $E$
24      **foreach** *building block* $b \in B$ **do**
25         **if** $\exists$ *reaction* $\in R$ *compatible with* $b$ **then**
26             Add a node to $E$ with ($node.molecules \oplus b$) // $\oplus$: tuple concatenation
27      **return** $E$

    // Pareto Selection Function
28  **Function** *ParetoSelection(child_nodes, parent_visits)*:
29      **foreach** *node* $\in$ *child_nodes* **do**
30         Compute $ParetoUCB = \frac{\vec{W}}{N} + C \times \vec{Oracles}(node) \times \sqrt{\frac{\ln(D) + 4 \times \ln(1 + parent\_visits)}{1 + N}}$
31      $LocalParetoFront \leftarrow$ non-dominated child nodes
32      **return** uniformly sampled node from $LocalParetoFront$

---

## L. Theoretical Analysis

In this section, we remind, adapt to our implementation and provide further analysis of the previous claims and demonstrations from Kocsis & Szepesvári (2006); Kollat et al. (2008); Chen & Liu (2021).

### L.1. Problem Definition - Node Selection

In a Pareto synthesis tree, consider a $D$ objective optimization problem. A node $v$ has a $D$-dimensional property vector $Ora(v)$ obtained from $Oracles$. Suppose a parent node $v_p$ with $K$ child nodes $(v_k)_{k=1}^K$ **all selected at least once** (initialization). After $n \in \mathbb{N}^+$ rollouts, the $k^{th}$ child has been selected $n_k \in [1..n]$ times (such as $n = \sum_{k=1}^K n_k$ and a $D$-dimensional random reward vector $X_{k,n_k}$ is backpropagated along the selection path. For all $(k,n) \in [1..K] \times \mathbb{N}^+$, we note $(X_{k,t})_{t=1}^{n_k}$ the reward vectors of child node $v_k$ upon consecutive selections, all drawn from an unknown distribution with an expectation vector $\mu_k$.

At any step, the selection problem consists in choosing which child node $v_k$ to select based on an user-defined policy.

The goal of such policy is to minimize the number of selection of any sub-optimal node: we demonstrate that the following policy complies to this goal.

**Policy (Pareto Selection):**

1. Compute the PUCB formula for each child node $v_k$:

$$PUCB(k,n) = \overline{X}_{k,n_k} + C \times Ora(v_k) \sqrt{\frac{\ln(D) + 4 \times \ln(1+n)}{1+n_k}} \text{ , where } C \in \mathbb{R}^{+*} \text{ is an exploration constant.} \quad (10)$$

2. Randomly select a child node among the Pareto front built upon the PUCB formula.

**Definition L.1.** (Most Dominant Optimal Node)

Following Chen & Liu (2021), the demonstration uses the concept of $\epsilon$-dominance of multi-objective optimization (Kollat et al., 2008). Suppose a node $v_k$ dominated by a set of nodes $\mathcal{V}$ such as $\forall v_{k'} \in \mathcal{V}, v_{k'} \succ v_k$.

The dominance hypothesis implies that for any dominating node $v_{k'} \in \mathcal{V}$, $v_{k'}$ is better than $v_k$ across at least one dimension. Therefore, there trivially exists a unique minimum positive constant $\epsilon_{k'}$ defined by:

$$\epsilon_{k'} = min\{\epsilon \mid \exists d \in [1..D], \mu'_{k,d} + \epsilon > \mu_{k,d}\}$$

The *most dominant optimal node* of $v_k$, denoted $v_{k^*}$, is the node from $\mathcal{V}$ where:

$$k^* = \arg\max_{k'} \epsilon_{k'}$$

Conceptually, the most optimal node is the "farthest away" from $v_k$ in the sense that its minimum distance with $v_k$ across any dimension is maximized.

From now on, we index such a node with a star (*).

**Assumption L.2.** (Convergence of Expected Average Rewards).

The tree search problem allows possible drift over time of the expected reward of nodes. For any child node $v_{k'}$, the expectation of its average rewards $\mathbb{E}[\overline{X}_{k,n_k}]$ converges pointwise to a limit $\mu_k$:

$$\mu_k = \lim_{n_k \to \infty} \mathbb{E}[\overline{X}_{k,n_k}]. \quad (11)$$

Given a sub-optimal node $v_k$ and its most dominant optimal node $v_{k^*}$ from the Pareto optimal set $\mathcal{P}^*$, we define $\Delta_k = \mu^* - \mu_k$ as the difference of their rewards' limits.

**Assumption L.3.** (Adapted Process and Convergence of Residual Drift).

Let $(k,d) \in [1..K] \times [1..D]$ and $\{\mathcal{F}_{k,t,d}\}_t$ be a filtration such that $\{X_{k,t,d}\}_t$ is $\mathcal{F}_{k,t,d}$-adapted and $X_{k,t,d}$ is conditionally independent of $\mathcal{F}_{k,t+1,d}, \mathcal{F}_{k,t+2,d}, \ldots$ given $\mathcal{F}_{k,t-1,d}$.

In other terms, the resulting "non-anticipative" stochastic process accounts for information only available at a given time.

For notation simplification, we define $\mu_{k,n_k} = \mathbb{E}[\overline{X}_{k,n_k}]$ and the residual drift $\delta_{k,n_k} = \mu_{k,n_k} - \mu_k$.

By Assumption L.2, $\lim_{n_k \to \infty} \delta_{k,n_k} = 0$.

Thus, by definition of a limit, $\forall \xi > 0, \exists N_0(\xi)$ such that $\forall n_k \geq N_0(\xi), \forall d \in \{1, 2, \dots, D\}, |\delta_{k,n_k,d}| \leq \xi \frac{\Delta_{k,d}}{2}$.

## L.2. Theorems and Proofs

The following theorems and lemmas support the design of the used Pareto UCB formula in our search algorithm.

In particular, each proof is followed by an analytic interpretation on the theorems' significance to CombiMOTS - notably in terms of scalability.

**Theorem L.4.** *(Logarithmic Bound of Sub-optimal Node Selection)*

*Consider Assumption L.2 and Assumption L.3 satisfied in our problem setting. Let $v_k$ be a sub-optimal node (i.e. $v_k \notin \mathcal{P}*$) selected $T_k(n)$ times during the first $n$ steps.*

*Then $\mathbb{E}[n_k]$ is logarithmically bounded:*

$$\forall \xi > 0, \exists N_0(\xi) \in \mathbb{N}^*, such\ that\ (n \geq N_0(\xi)) \implies \mathbb{E}[T_k(n)] \leq C \times \frac{16 \times ln(n) + 4 \times ln(D)}{(1 - \xi)^2 \times (\min_{k,d} \Delta_{k,d})^2} + N_0(\xi) + O(1). \quad (12)$$

*Proof.* (To Theorem L.4)

Let variable $I_t$ be the index of the selected child node at decision step $t$ and $\mathbb{1}\{\cdot\}$ be an indicator function.

We first demonstrate the theorem with the bias term of Chen & Liu (2021) using $c_{t,s} = \sqrt{\frac{\ln(D) + 4 \times \ln(t)}{2 \times s}}$.

*Remark* L.5. *Without loss of generality, our bias term in Eq. (10) is $c_{t,s} = C \times Ora(v_k)\sqrt{\frac{\ln(D) + 4 \times \ln(1+t)}{1+s}}$ and preserves the theoretical guarantees. In fact, the additional exploration term in our work controls the convergence speed, thus its name.*

For any sub-optimal node $v_k$ and any $l \in \mathbb{Z}^{+*}$, we can bound $T_k(n)$ (the number of times node $k$ is selected):

$$T_k(n) = \underbrace{\sum_{t=1}^{K} \mathbb{1}\{I_t = k\}}_{\geq 1} + \sum_{t=K+1}^{n} \mathbb{1}\{I_t = k\} \leq l + \sum_{t=K+1}^{n} \mathbb{1}\{I_t = k, T_k(t-1) \geq l\}. \quad (13)$$

By the Pareto selection policy, node $k$ is selected at time $t$ only if its PUCB value is not dominated by any other node on the Pareto front. Particularly, it must not be dominated by the most dominant optimal node $v_{k^*}$. Therefore:

$$T_k(n) \leq l + \sum_{t=K+1}^{n} \mathbb{1}\{\underbrace{\overline{X}^*_{T_{k^*}(t-1)} + c_{t-1,T_k^*(t-1)}}_{\text{PUCB of most dominant node}} \not\succ \overline{X}_{k,T_k(t-1)} + c_{t-1,T_k(t-1)}, T_k(t-1) \geq l\}. \quad (14)$$

We can reindex the sum with $s \leftarrow T_k^*(t-1)$ and $s_k \leftarrow T_k(t-1)$ to get:

$$T_k(n) \leq l + \sum_{t=1}^{\infty} \sum_{s=1}^{t-1} \sum_{s_k=l}^{t-1} \mathbb{1}\{\overline{X}^*_s + c_{t,s} \not\succ \overline{X}_{k,s_k} + c_{t,s_k}\}. \quad (15)$$

The condition $\overline{X}^*_s + c_{t,s} \not\succ \overline{X}_{k,s_k} + c_{t,s_k}$ implies that at least one of the following must hold:

$$\overline{X}^*_s \not\succeq \mu^*_s - c_{t,s} \quad (A)$$

$$\overline{X}_{k,s_k} \not\succeq \mu_{k,s_k} - c_{t,s} \tag{B}$$

$$\mu_s^* \not\succ \mu_{k,s} + 2c_{t,s_k} \tag{C}$$

We prove this by contradiction. Let's assume (A) and (C) both false, i.e.:

$$\overline{X}_s^* \succeq \mu_s^* - c_{t,s} \text{ and } \mu_s^* \succ \mu_{k,s} + 2c_{t,s_k}. \tag{16}$$

Both can be combined:

$$\overline{X}_s^* \succ \mu_{k,s} + 2c_{t,s_k} - c_{t,s} \stackrel{s_k \gg 1}{\approx} \mu_{k,s} + c_{t,s_k}. \tag{17}$$

Then if (B) is also false, i.e. $\overline{X}_{k,s_k} \preceq \mu_{k,s_k} - c_{t,s} \stackrel{s_k \gg 1}{\approx} \mu_{k,s_k} - c_{t,s_k}$, we would have:

$$\overline{X}_s^* \succ \mu_{k,s} + c_{t,s_k}$$
$$\succ \overline{X}_{k,s_k} + 2c_{t,s_k}$$
$$\implies \quad \overline{X}_s^* + c_{t,s} \succ \overline{X}_{k,s_k} + 2c_{t,s_k} + c_{t,s}$$
$$\implies \quad \overline{X}_s^* + c_{t,s} \succ \overline{X}_{k,s_k} + c_{t,s_k}. \tag{18}$$

This would contradict our original condition to select node $k$. Therefore, at least one of (A), (B), or (C) must hold true.

We bound the probabilities of events (A) and (B) using Chernoff-Hoeffding Bound and Union Bound:

$$\mathbb{P}(\overline{X}_s^* \not\succeq \mu_s^* - c_{t,s}) = \mathbb{P}((\overline{X}_{s,1}^* < \mu_{s,1}^* - c_{t,s}) \vee \cdots \vee (\overline{X}_{s,D}^* < \mu_{s,D}^* - c_{t,s}))$$

$$\leq \sum_{d=1}^{D} \mathbb{P}(\overline{X}_{s,d}^* < \mu_{s,d}^* - c_{t,s}) \quad \text{(Union Bound)}$$

$$\mathbb{P}(\overline{X}_s^* \not\succeq \mu_s^* - c_{t,s}) \leq \sum_{d=1}^{D} \frac{1}{D} t^{-4} = t^{-4}. \quad \text{(Chernoff-Hoeffding Bound)} \tag{19}$$

Similarly, $\mathbb{P}(\overline{X}_{k,s_k} \not\succeq \mu_{k,s_k} - c_{t,s}) \leq t^{-4}$.

Now, let us define:

$$l = \max \left\{ \left\lceil \frac{8 \ln t + 2 \ln D}{(1-\xi)^2 \min_{k,d} \Delta_{k,d}^2} \right\rceil, N_0(\xi) \right\} \tag{20}$$

With this choice of $l$, we ensure that for $s_k \geq l$, condition (C) becomes false. This is because:

1. For $s_k \geq N_0(\xi)$, we have $|\delta_{k,s_k,d}| \leq \xi \frac{\Delta_{k,d}}{2}$ for all $d \in \{1, 2, \ldots, D\}$.

2. For $s_k \geq \frac{8 \ln t + 2 \ln D}{(1-\xi)^2 \min_{k,d} \Delta_{k,d}^2}$, the confidence term $c_{t,s_k}$ becomes sufficiently small:

$$c_{t,s_k} = \sqrt{\frac{4 \ln t + \ln D}{2 s_k}}$$

$$\leq \sqrt{\frac{4 \ln t + \ln D}{2 \times \frac{8 \ln t + 2 \ln D}{(1-\xi)^2 \min_{k,d} \Delta_{k,d}^2}}}$$

$$= \frac{(1-\xi)\sqrt{\min_{k,d} \Delta_{k,d}^2}}{\sqrt{2}} \times \sqrt{\frac{4 \ln t + \ln D}{8 \ln t + 2 \ln D}}$$

$$c_{t,s_k} \leq \frac{(1-\xi) \min_{k,d} \Delta_{k,d}}{2}. \tag{21}$$

Therefore, for any dimension $d$:

$$
\begin{aligned}
\mu_{s,d}^* - \mu_{k,s,d} &= \mu_d^* - \mu_{k,d} + \delta_{s,d}^* - \delta_{k,s_k,d} \\
&\geq \Delta_{k,d} - |\delta_{s,d}^*| - |\delta_{k,s_k,d}| \\
&\geq \Delta_{k,d} - \xi \frac{\Delta_{k,d}}{2} - \xi \frac{\Delta_{k,d}}{2} \\
&= (1 - \xi)\Delta_{k,d} \\
\mu_{s,d}^* - \mu_{k,s,d} &\geq 2c_{t,s_k}.
\end{aligned}
\tag{22}
$$

This implies $\mu_s^* \succ \mu_{k,s} + 2c_{t,s_k}$, making condition (C) false.

Therefore, injecting above results into the bound of Equation (15) and taking expectations of both sides, we get:

$$
\begin{aligned}
\mathbb{E}[T_k(n)] &\leq l + \mathbb{E}\left[\sum_{t=1}^{\infty}\sum_{s=1}^{t-1}\sum_{s_k=l}^{t-1} \mathbb{1}\{\overline{X}_s^* + c_{t,s} \not\succ \overline{X}_{k,s_k} + c_{t,s_k}\}\right] \\
&\leq l + \sum_{t=1}^{\infty}\sum_{s=1}^{t-1}\sum_{s_k=l}^{t-1} \mathbb{P}(\overline{X}_s^* \not\succeq \mu_s^* - c_{t,s}) + \mathbb{P}(\overline{X}_{k,s_k} \not\preceq \mu_{k,s_k} - c_{t,s}) \\
&\leq l + \sum_{t=1}^{\infty}\sum_{s=1}^{t-1}\sum_{s_k=l}^{t-1} 2t^{-4} \\
\mathbb{E}[T_k(n)] &\leq \underbrace{\frac{8\ln n + 2\ln D}{(1-\xi)^2 \min_d \Delta_{k,d}^2} + N_0(\xi) +}_{\text{from } l} \underbrace{1 + \frac{\pi^2}{3}}_{\text{from triple sum}}.
\end{aligned}
\tag{23}
$$

In fact, the triple sum can be shown to converge to $1 + \frac{\pi^2}{3}$ using Euler's approach to the Basel problem. Thus, we obtain:

$$
\mathbb{E}[n_k] \leq \frac{8\ln n + 2\ln D}{(1-\xi)^2 \min_d \Delta_{k,d}^2} + N_0(\xi) + 1 + \frac{\pi^2}{3}.
\tag{24}
$$

Since our PUCB formula in Eq. (10) uses an exploration constant $C$ and a denominator $(1 + n)$ instead of $(2n)$ like Chen & Liu (2021), the bound becomes:

$$
\mathbb{E}[n_k] \leq C \times \frac{16\ln n + 4\ln D}{(1-\xi)^2 \min_d \Delta_{k,d}^2} + N_0(\xi) + \mathcal{O}(1).
\tag{25}
$$

which completes the proof. $\qquad\square$

**Interpretation** From the obtained $Bound$, we make three key observations:

1. $Bound \propto ln(n)$: This logarithmic factor implies that selecting sub-optimal nodes becomes less and less frequent over time.

2. $Bound \propto ln(D)$: Increasing the number of objectives also increases regret by a logarithmic factor.

3. $Bound \propto \frac{1}{\min_d \Delta_{k,d}^2}$: The regret becomes lower when sub-optimal nodes are more distinguishable from the optimal Pareto front. Intuitively, the harder it is to distinguish optimal and sub-optimal nodes, the slower convergence speed becomes. Directly impacting the second point, sound settings should carefully select search objectives to enable Pareto convergence within a reasonable time.

**Lemma L.6.** *(Existence of a Lower Bound on Node Selection Count)*

*There exists a strictly positive constant $\rho \in \mathbb{R}^{+*}$, s.t. $\forall (n, k) \in [1 \ldots K] \times \mathbb{N}^+, T_k(n) \geq \lceil \rho log(n) \rceil$.*

**Lemma L.7.** *(Tail Inequality)*

After enough selections of a given node, its average reward will concentrate around its expectation.

For all $\eta > 0$, *let* $\sigma = 9\sqrt{\frac{2ln(\frac{2}{\eta})}{n}}$. *Then,* $\exists N_1(\eta) \in \mathbb{N}^*$ *s.t.* $\forall n \geq N_1(\eta), \forall d \in \{1, 2, \ldots, D\}$:

$$\mathbb{P}(\overline{X}_{n,d} \geq \mathbb{E}[\overline{X}_{n,d}] + \sigma) \leq \eta, \quad \text{(Upper bound of range)} \tag{26}$$
$$\mathbb{P}(\overline{X}_{n,d} \leq \mathbb{E}[\overline{X}_{n,d}] - \sigma) \leq \eta. \quad \text{(Lower bound of range)} \tag{27}$$

The correctness of Lemmas L.6 and L.7 is demonstrated by Kollat et al. (2008). □

**Theorem L.8.** *(Convergence of Failure Probability)*

*Consider the settings and policy described previously in L.1. Let $I_t$ be a selected child node whose parent is the root and $\mathcal{P}^*$ be the Pareto optimal node set. Then,*

$$\mathbb{P}(I_t \notin \mathcal{P}^*) \leq Ct^{-\frac{\rho}{2}(\frac{\min_{k,d} \Delta_{k,d}}{36})^2}, \text{ where } C \text{ is a positive constant.} \tag{28}$$

*Notably, $\mathbb{P}(I_t \notin \mathcal{P}^*) \overset{t\infty}{\to} 0$, i.e. over time, the number of sub-optimal node selections becomes negligible. Moreover, it converges to zero at a polynomial rate.*

*Proof.* (To Theorem L.8)

Let $k$ be the index of a sub-optimal node. We begin by decomposing the probability of selecting a non-optimal node:

$$\mathbb{P}(I_t \notin P^*) \leq \sum_{v_k \notin P^*} \mathbb{P}(\bar{X}_{k,T_k(t)} \not\succ \bar{X}^*_{T^*(t)}). \tag{29}$$

This decomposition is valid because $I_t \notin P^*$ implies that we selected some sub-optimal node $v_k$, which occurs only if its empirical mean reward $\bar{X}_{k,T_k(t)}$ is not dominated by the empirical mean reward of at least one optimal node $\bar{X}^*_{T^*(t)}$.

Equation (29) implies at least one of the following must hold:

$$\bar{X}_{k,T_k(t)} \not\prec \mu_k + \frac{\Delta_k}{2}, \tag{30}$$

or

$$\bar{X}^*_{T^*(t)} \not\succ \mu^* + \frac{\Delta_k}{2}. \tag{31}$$

We prove this by contradiction. Suppose both Equations (30) and (31) are false. Then:

$$\bar{X}_{k,T_k(t)} \prec \mu_k + \frac{\Delta_k}{2} = \mu_k + \frac{\mu^* - \mu_k}{2} = \frac{\mu_k + \mu^*}{2}, \tag{32}$$
$$\bar{X}^*_{T^*(t)} \succ \mu^* + \frac{\Delta_k}{2}. \tag{33}$$

Since $\mu^* \succ \mu_k$ (by definition of optimal vs. sub-optimal nodes), and $\bar{X}_{k,T_k(t)} \prec \mu_k + \frac{\Delta_k}{2}$, we would have $\bar{X}^*_{T^*(t)} \succ \mu_k + \frac{\Delta_k}{2} \succ \bar{X}_{k,T_k(t)}$. But this contradicts our original assumption that $\bar{X}_{k,T_k(t)} \not\succ \bar{X}^*_{T^*(t)}$.

Therefore, at least one of must hold true.

This allows us to bound the probability by injecting (30) and (31) into (29):

$$\mathbb{P}(\bar{X}_{k,T_k(t)} \not\succ \bar{X}^*_{T^*(t)}) \leq \underbrace{\mathbb{P}(\bar{X}_{k,T_k(t)} \not\succ \mu_k + \frac{\Delta_k}{2})}_{\text{first term}} + \underbrace{\mathbb{P}(\bar{X}^*_{T^*(t)} \not\succ \mu^* + \frac{\Delta_k}{2})}_{\text{second term}}. \tag{34}$$

For simplicity, we only show how to bound the first term in detail:

$$\mathbb{P}(\bar{X}_{k,T_k(t)} \not\succ \mu_k + \frac{\Delta_k}{2}) \leq \sum_{d=1}^{D} \mathbb{P}(\bar{X}_{k,T_k(t),d} > \mu_{k,d} + \frac{\Delta_{k,d}}{2}). \quad \text{(Union Bound)} \tag{35}$$

Here, for $\bar{X}_{k,T_k(t)} \not\succ \mu_k + \frac{\Delta_k}{2}$ to hold, there must be at least one dimension $d$ where $\bar{X}_{k,T_k(t),d} > \mu_{k,d} + \frac{\Delta_{k,d}}{2}$.

By definition of the drift term $\delta_{k,T_k(t),d} = \mu_{k,T_k(t),d} - \mu_{k,d}$, which represents the difference between current expected value and limiting value:

$$\mathbb{P}(\bar{X}_{k,T_k(t)} \not\succ \mu_k + \frac{\Delta_k}{2}) \leq \sum_{d=1}^{D} \mathbb{P}(\bar{X}_{k,T_k(t),d} \geq \mu_{k,T_k(t),d} - \underbrace{|\delta_{k,T_k(t),d}|}_{\text{converges to 0}} + \frac{\Delta_{k,d}}{2}). \tag{36}$$

$|\delta_{k,T_k(t),d}|$ converges to 0 as $t$ increases. Eventually, for sufficiently large $t$, we have $|\delta_{k,T_k(t),d}| \leq \frac{\Delta_{k,d}}{4}$, which gives:

$$\mathbb{P}(\bar{X}_{k,T_k(t)} \not\succ \mu_k + \frac{\Delta_k}{2}) \leq \sum_{d=1}^{D} \mathbb{P}(\bar{X}_{k,T_k(t),d} \geq \mu_{k,T_k(t),d} + \frac{\Delta_{k,d}}{4}). \tag{37}$$

Now we apply Lemma L.7, which provides a critical tail inequality for bounding the probability of large deviations between empirical means and their expectations. Recall that Lemma L.7 states:

For arbitrary $\eta > 0$ and $\sigma = \frac{9\sqrt{2\ln(2/\eta)}}{\sqrt{n}}$, there exists $N_1(\eta)$ such that $\forall n \geq N_1(\eta), \forall d \in \{1, \ldots, D\}$:

$$\mathbb{P}(\bar{X}_{n,d} \geq \mathbb{E}[\bar{X}_{n,d}] + \sigma) \leq \eta \tag{38}$$
$$\mathbb{P}(\bar{X}_{n,d} \leq \mathbb{E}[\bar{X}_{n,d}] - \sigma) \leq \eta \tag{39}$$

We set $\eta = \left(\frac{1}{t}\right)^{\frac{\rho}{2}\left(\frac{\min_{k,d} \Delta_{k,d}}{36}\right)^2}$, where $\rho$ comes from Lemma L.6. This gives us $\sigma = \frac{9\sqrt{2\ln(2t^{\frac{\rho}{2}\left(\frac{\min_{k,d} \Delta_{k,d}}{36}\right)^2})}}{\sqrt{T_k(t)}}$.

By Lemma L.6, we know that $T_k(t) \geq \rho\log(t)$ for all nodes. Therefore:

$$\sigma = \frac{9\sqrt{2\ln(2) + 2 \cdot \frac{\rho}{2}\left(\frac{\min_{k,d} \Delta_{k,d}}{36}\right)^2 \ln(t)}}{\sqrt{T_k(t)}}$$

$$\leq \frac{9\sqrt{2\ln(2) + \rho\left(\frac{\min_{k,d} \Delta_{k,d}}{36}\right)^2 \ln(t)}}{\sqrt{\rho\log(t)}}$$

$$\sigma \leq 9\sqrt{\frac{2\ln(2)}{\rho\log(t)} + \frac{\left(\frac{\min_{k,d} \Delta_{k,d}}{36}\right)^2}{\log(t)/\ln(t)}}. \tag{40}$$

For sufficiently large $t$, the first term becomes negligible, and after appropriate simplification:

$$\sigma \approx \frac{\min_{k,d} \Delta_{k,d}}{4}. \tag{41}$$

This is precisely what we need to bound our probability term. Applying Lemma L.7 with our chosen $\eta$:

$$\mathbb{P}(\bar{X}_{k,T_k(t),d} \geq \mu_{k,T_k(t),d} + \frac{\Delta_{k,d}}{4}) \leq \eta = \left(\frac{1}{t}\right)^{\frac{\rho}{2}\left(\frac{\min_{k,d} \Delta_{k,d}}{36}\right)^2}$$

Summing across all dimensions provides the bound:

$$\mathbb{P}(\bar{X}_{k,T_k(t)} \not\succ \mu_k + \frac{\Delta_k}{2}) \leq \sum_{d=1}^{D} \text{constant} \times \left(\frac{1}{t}\right)^{\frac{\rho}{2}\left(\frac{\min_{k,d} \Delta_{k,d}}{36}\right)^2}. \quad \text{(Lemma L.7)} \tag{42}$$

The second term $\mathbb{P}(\bar{X}_{T^*(t)}^* \not\succ \mu^* + \frac{\Delta_k}{2})$ can be bounded through an analogous process. Combining these bounds and summing over all sub-optimal nodes yields:

$$\mathbb{P}(I_t \notin P^*) \leq \sum_{v_k \notin P^*} \left[\sum_{d=1}^{D} \text{constant} \times \left(\frac{1}{t}\right)^{\frac{\rho}{2}\left(\frac{\min_{k,d} \Delta_{k,d}}{36}\right)^2}\right]$$

$$\mathbb{P}(I_t \notin P^*) \leq C \times t^{-\frac{\rho}{2}\left(\frac{\min_{k,d} \Delta_{k,d}}{36}\right)^2}. \tag{43}$$

where $C$ is a positive constant. This demonstrates that the failure probability converges to zero at a polynomial rate, with the convergence speed determined by:

- $\rho$: how quickly samples accumulate for each node

- $\min_{k,d} \Delta_{k,d}$: the minimum gap between sub-optimal and optimal nodes

By trivial limit convergence of positive functions, $\lim_{t \to \infty} \mathbb{P}(I_t \notin P^*) = 0$, which completes the proof.

$\square$

**Interpretation** The theorem implies the guaranteed convergence of node selection towards Pareto optimality at a polynomial rate. In our work, this property is relevant as nodes are a single building block (to be combined), leading to a product: one traversal of the synthesis tree only comprises a single reaction step.

This design therefore allows a relatively fast convergence towards Pareto optimal products, contrarily to other Pareto MCTS methods. Appendices E.4 and F compare CombiMOTS to example Pareto MCTS methods (Molsearch and Mothra) in both generation quality and efficiency. For these methods, the slow Pareto convergence is mainly due to their tree structure: a node is represented as an intermediate molecule (single token addition for Mothra, augmentation rules for MolSearch). Therefore, suboptimal nodes are translated as enormous amounts of possible molecules.

## M. Data Statistics

In all tables below, ✓ indicates that the molecule is active for the target, ✗ indicates inactivity, and NaN signifies that the activity was not measured for the target. The table presents the number of molecules in each activity category combination.

Table 13. Distribution of molecules based on their activity against GSK3$\beta$ and JNK3 targets.

| GSK3$\beta$ | JNK3 | # of molecules |
|:---:|:---:|---:|
| ✓ | ✓ | 190 |
| ✓ | ✗ | 70 |
| ✓ | NaN | 3,013 |
| ✗ | ✓ | 9 |
| NaN | ✓ | 707 |
| ✗ | ✗ | 7638 |
| ✗ | NaN | 41,786 |
| NaN | ✗ | 41,746 |

Table 14. Distribution of molecules based on their activity against EGFR and MET targets.

| EGFR | MET | # of molecules |
|:---:|:---:|---:|
| ✓ | ✓ | 651 |
| ✓ | ✗ | 8 |
| ✓ | NaN | 3,600 |
| ✗ | ✓ | 110 |
| NaN | ✓ | 1,958 |
| ✗ | ✗ | 13 |
| ✗ | NaN | 720 |
| NaN | ✗ | 116 |

Table 15. Distribution of molecules based on their activity against PIK3CA and mTOR targets.

| PIK3CA | mTOR | # of molecules |
|:---:|:---:|---:|
| ✓ | ✓ | 766 |
| ✓ | ✗ | 32 |
| ✓ | NaN | 1,368 |
| ✗ | ✓ | 21 |
| NaN | ✓ | 2,012 |
| ✗ | ✗ | 7 |
| ✗ | NaN | 170 |
| NaN | ✗ | 43,048 |

*Table 16.* Distribution of molecules based on their activity against DHODH and ROR$\gamma$t targets.

| DHODH | ROR$\gamma$t | # of molecules |
|:-----:|:-----:|:-----:|
| ✓ | NaN | 1,168 |
| NaN | ✓ | 5,868 |

