# OpenReview forum: "CombiMOTS: Combinatorial Multi-Objective Tree Search for Dual-Target Molecule Generation"
_ICML.cc/2025/Conference — ICML 2025 poster_

### Official Review · Reviewer_KAF5 · 2025-03-12

**Overall Recommendation:** 3

**Summary:**

The paper introduces CombiMOTS - a Pareto Monte Carlo Tree Search based approach that is designed to efficiently handle complex multi-objective optimization problems. In particular, the authors tackle the dual-targeting molecule generation problem and evaluate the framework using three protein pairs. For this, they narrow down a large molecule fragment space into a synthesizable one and generate molecules with optimization of docking scores against two proteins.

**Claims And Evidence:**

1. The paper provides strong evidence that CombiMOTS generates molecules with better docking scores, diversity, and novelty. However, synthetic accessibility is not discussed in detail. The authors report SA scores, which are calculated based on the molecular structures, but do not include a retrosynthesis validation study to check if the proposed molecules can truly be synthesized.

2. The authors demonstrate that CombiMOTS consistently outperforms existing methods, however it seems that several important baselines are missing. Additionally, it is not quite clear how the proposed approach compares to baselines in terms of efficiency.

**Essential References Not Discussed:**

FREED++ framework [1] was not cited, although it performs fragment-based molecule generation using RL and was reported to outperform REINVENT in certain scenarios. There are several models for multi-objective drug optimization that use MCTS [2-5] which were not cited. The authors should also consider adding these models as baselines. There is also one recent paper that introduces Pareto optimization for drug design [6].

[1] Telepov, Alexander, et al. "FREED++: Improving RL Agents for Fragment-Based Molecule Generation by Thorough Reproduction." arXiv preprint arXiv:2401.09840 (2024).
[2] Suzuki, Takamasa, et al. "Mothra: Multiobjective de novo molecular generation using monte carlo tree search." Journal of Chemical Information and Modeling 64.19 (2024): 7291-7302.
[3] Roucairol, Milo, et al. "DrugSynthMC: An Atom-Based Generation of Drug-like Molecules with Monte Carlo Search." Journal of Chemical Information and Modeling 64.18 (2024): 7097-7107.
[4] Sun, Mengying, et al. "Molsearch: search-based multi-objective molecular generation and property optimization." Proceedings of the 28th ACM SIGKDD conference on knowledge discovery and data mining. 2022.
[5] Qian, Hao, et al. "AlphaDrug: protein target specific de novo molecular generation." PNAS nexus 1.4 (2022): pgac227.
[6] Yang, Yaodong, et al. "Enabling target-aware molecule generation to follow multi objectives with Pareto MCTS." Communications Biology 7.1 (2024): 1074.

**Experimental Designs Or Analyses:**

1. In Table 1 error rates are missing. It would be interesting to see how consistent CombiMOTS is and how it compares to other models in terms of stability of metrics.
2. The authors evaluate their approach in terms of toxicity of generated molecules. It would be more informative to evaluate the impact of toxicity optimization in an ablation study.
3. Although synthetic accessibility score is a common metric to estimate synthetic complexity of a compound, there are more advanced metrics capturing not only structural features of a molecule, but also possible retrosynthetic pathways (e.g., BR-SAscore and RAscore). Authors should consider calculating these metrics if they claim that synthetic accessibility of generated molecules increases.

**Methods And Evaluation Criteria:**

The proposed CombiMOTS method, real-world cases and evaluation criteria (originality and quality) do make sense for the problem of dual-targeting molecule generation. However, it is not quite evident how the proposed methodology compares to other existing approaches.

**Other Comments Or Suggestions:**

1. The experiment reported in section C.1.1 of the Appendix demonstrates that Pareto optimization leads to better-balanced trade-offs than a naive scalarized reward function. It would be also interesting to investigate how adding Pareto optimization impacts the computational costs required to generate molecules.
2. Comparison of CombiMOTS to baselines in terms of time was not reported, although given the complexity of multi-objective drug optimization, it would be interesting to see how much time is required for the proposed approach to achieve a certain quality of molecules.
3. Exploring the aspect of selective molecule generation could be beneficial, as technically the idea remains the same, while expanding the scope of the study. Additional case studies to this point would strengthen the work by better demonstrating its practical utility.

**Other Strengths And Weaknesses:**

Strengths:
1. The research in generation of dual-targeting molecules is topical and important for the broad scientific community.
2. The paper is easy to follow. The plots and figures are visually attractive.

Weaknesses:
1. It seems that the idea of using Pareto MCTS for molecule generation is not novel. Although generating dual-targeting molecules is a more complex task, it is very similar to the conventional multi-objective molecule optimization.
2. Some aspects of experimental design and analyses need improvements, e.g., inclusion of more baselines and experimental studies on robustness and efficiency of the presented approach.
3. Not all the relevant papers on this topic were cited suggesting that the authors might not be aware of many significant developments in the field.
4. Some claims are not supported with clear evidence or just vague  (e.g., about the proposed method generating interpretable compounds or the better synthetic accessibility of the generated molecules).

**Questions For Authors:**

1. Why were only the three baselines considered? Why not consider other SOTA models that use MCTS, Pareto optimization and/or diffusion?
2. The reported 11s/rollout time means that full runs take days. How does this compare to the other methods in terms of efficiency?
3. What does “interpretable” compounds in line 96 mean? What are the statistically solid results supporting that claim?
4. Is it possible to provide a retrosynthetic evaluation of the generated molecules?
5. Is it possible to calculate standard deviations for metrics in Table 1 (e.g., after several runs with different random seeds)?
6. Is it possible to compare CombiMOTS with and without toxicity optimization to see the differences more clearly?
7. Is it possible to compare CombiMOTS with baselines in terms of time?

**Relation To Broader Scientific Literature:**

The issue of multi-objective fragment-based molecular optimization has been investigated in various studies. The use of Pareto MCTS for molecular generation was studied in [1]. The novelty of the proposed approach lies in the application of Pareto MCTS for dual-targeting molecule generation.

[1] Yang, Yaodong, et al. "Enabling target-aware molecule generation to follow multi objectives with Pareto MCTS." Communications Biology 7.1 (2024): 1074.

**Theoretical Claims:**

Not applicable.

---

> ### Author Rebuttal · Authors · 2025-04-01
>
> Dear reviewer **KAF5**, thank you for your constructive suggestions. We address your concerns below and through https://anonymous.4open.science/r/CombiMOTS-0FEB.
>
> Concerns
> -
>
> ### 4. Synthesizability Metrics and Retrosynthetic Evaluation
>
> We assess CombiMOTS's synthesizability against baselines to justify using Enamine rather than post-hoc evaluation.
>
> Synthesizability metrics can be:
> - **Rule-based** (e.g., SAscore): Efficient but fail to capture real-world constraints.
> - **Retrosynthetic-based**: Recursively breaking down a target into industry-available precursors. However, it is expensive and limited to reference stocks (e.g.,Aizynthfinder [1] on the ZINC database). We use the opposite workflow (from precursors, find optimal solutions within bounded iterations - see reply `H` to **pmFQ**)
> - **Learning-based** (e.g., RAscore, BR-SAscore): Context-adaptable but data-dependent.
> > Note: Enamine’s 80% synthesis success rate and SyntheMol’s experimental validations [2] motivate practical synthesis over mere metric evaluation.
>
> #### Comparative Analysis
> - **Density Distributions:**
>   Density distributions (in repo) include:
>   - Generated molecules (all models)
>   - Enamine building blocks (expects high synthesizability)
>   - COMPAS-3 dataset [3] (contains hard-to-synthesize polycyclic compounds)
> - **Observations:**
>   - COMPAS-3 is deemed synthesizable by SA & RAscore but not by BR-SAscore.
>   - Enamine excels on RAscore but performs similarly or worse on other metrics.
>   - MARS unrealistically outperforms Enamine on SA/BR-SAscore.
>   - CombiMOTS outperforms baselines on RAscore but aligns with/worsens on others.
> #### Aizynthfinder [1] Retrosynthetic Study
> Among 1544 CombiMOTS molecules, 162 (10.42%) are not solved by Aizynthfinder—due to its ZINC dependency (example in repo).
> These inconsistencies among metrics underscore why leveraging available precursor data is relevant to our motives.
>
> >B) Selective molecule generation
>
> We explore a workflow for CDK7 inhibitors as a case study. Off-targets (CDK1, 2, 9, 12, 13) are reported in [4,5].
>
> CombiMOTS could be adapted by:
> - Searching compounds maximizing CDK7 activity and minimizing off-target activity.
> - Narrow the search space on known active scaffolds/warheads.
> - Post-hoc filtering (medicinal chemistry, in-silico validation). While preliminary and beyond the scope of our current work, repository statistics summarize this workflow.
>
> Other Questions
> -
> >1. Additional Baselines
>
> We thank the reviewer for suggesting references to integrate in future versions. We note distinctions with CombiMOTS:
> - DrugSynthMC [6] generates drug-like molecules using n-grams but lacks support for protein target setting.
> - ParetoDrug [7] extends PMCTS to protein-specific generation via atom-based, Lmser transformer models. Though cited, this method remains single-target (no multi-target code).
> - Mothra [8] is PMCTS-based but limited to single-target RNN decoding.
> - MolSearch [9] tackles on dual-target optimization rather than de novo generation by combining PMCTS with fragment-based "design moves".
>
> We adapt Mothra and MolSearch to the GSK3B-JNK3 task. Repository metrics/plots show CombiMOTS's superiority across all metrics, likely due to:
> - Mothra relying on single token additions.
> - MolSearch's dependence on learned augmentation rules
>     - Both require extensive sampling for Pareto convergence (see reply `H` to **pmFQ**).
> - In contrast, CombiMOTS's node structure leads to products in a single reaction, leading to faster convergence.
>
> >2. Efficiency & 7. Runtime
>
> One CombiMOTS tree traversal runs in $O(1)$ time, with hash table lookups (precomputed scores) being the main overhead when creating child nodes. However, oracles are called whenever a product is found: the computational overhead is bounded by the slower operation between $O(n_{children})$ and $\text{Oracles(product)}$.
>
> Runtimes of all evaluated models are in the repository. Notably, competing PMCTS methods are significantly slower.
>
> >3. Interpretable compounds.
>
> In Fragment-Based Drug Design, it means explaining molecular property through the fragments it is built upon. Here, FGIB helps find more dual-actives (see reply `4` to **6Myy**)
> >5. Error rates of Table 1.
>
> We update Table 1 with error rates, p-values, and separate evaluation of baselines for dual-active generation. CombiMOTS, as a search algorithm, returns all found products, making comparisons with generative models irrelevant.
>
> >6. Toxicity Optimization
>
> We add plots in repository for the corresponding ablation study to conclude on CombiMOTS's ability to optimize non-toxicity.
>
> Reference
> -
> [1] Genheden et al., *J. Cheminform.*, 2020
>
> [2] Swanson et al., *Nat. Mach. Intell.*, 2024
>
> [3] Wahab and Gershoni-Porane, *PCCP*, 2024
>
> [4] Constantin et al., *Br. J. Cancer*, 2023
>
> [5] Olson et al., *Cell Chem. Biol.*, 2019
>
> [6] Roucairol et al., *J. Cheminform.*, 2024
>
> [7] Yang et al., *Commun. Biol.*, 2024
>
> [8] Suzuki et al., *JCIM*, 2024
>
> [9] Sun et al., *KDD'22*, 2022

---

> > ### Comment · Reviewer_KAF5 · 2025-04-04
> >
> > I appreciate the authors making every attempt to address all the questions and concerns I had expressed. Thanks for the additional experimental results you included in the repository. While I find it difficult to critically assess all the new evidence given the time constraints, I'm inclined to raise my score. I do not have any further questions.

---

> > > ### Author Response · Authors · 2025-04-07
> > >
> > > Dear reviewer **KAF5**, we would like to express once again our gratitude towards your great advice and observations during the entire review process !
> > >
> > > Including answers to your concerns in future versions will surely improve our work's quality and standpoint on the topic. We greatly appreciate how you helped us underline multiple aspects of our method such as its relevance on synthesizability, how it compares against PMCTS baselines, its runtime or how it could be applied to other tasks.
> > >
> > > We also thank you for increasing your score!

---

### Official Review · Reviewer_pmFQ · 2025-03-13

**Overall Recommendation:** 3

**Summary:**

The paper proposes CombiMOTS, a combinatorial multi-objective tree search framework for dual-target molecule generation. It integrates Pareto Monte Carlo Tree Search (PMCTS) with fragment-based synthesis-aware generation. Key contributions include: (1) A reduced synthesizable fragment space via target-aware building blocks; (2) A Pareto-optimized MCTS algorithm for balancing conflicting objectives (target affinity, QED, SA); (3) Empirical validation showing superior performance in generating diverse, synthesizable molecules with balanced properties compared to baselines.

**Claims And Evidence:**

Claims about "superior diversity" (Table 1) lack statistical significance tests (e.g., p-values).

Toxicity prediction (Section 6.2) uses an ensemble of Chemprop models trained on a small dataset (1,478 molecules), risking overfitting.

**Essential References Not Discussed:**

Pareto optimization in drug design: Medina-Franco et al. (2013) proposed multi-target optimization frameworks.

Multi-objective RL: Yang et al. (2024a) applied Pareto MCTS to molecule generation but is uncited.

**Experimental Designs Or Analyses:**

Concern: Baseline implementations (e.g., REINVENT) use different objectives (QED-only vs. docking scores), making comparisons unfair.

Issue: Computational cost of CombiMOTS (25s/rollout for 6 objectives) is not compared to baselines.

**Methods And Evaluation Criteria:**

Use of industry-ready Enamine REAL Space ensures practical synthesizability. Pareto optimization aligns with multi-objective drug design.
However, docking scores are prioritized over QED/SA without justification. This may bias results toward unrealistic molecules.

**Other Comments Or Suggestions:**

Page 2: "Fromer & Coley, 2023; Luukkonen & Maagdenberg, 2023" lack full references.

Algorithm 1: Line 16 uses "max(v, selected.P)"—unclear if "max" applies to vectors.

**Other Strengths And Weaknesses:**

Weaknesses: Methods are well-described, but theoretical justification for Pareto UCB is unclear.

**Questions For Authors:**

How does the Pareto UCB formula ensure convergence to Pareto-optimal solutions? A theoretical analysis is needed.

Why prioritize docking scores over QED/SA in objectives? Would including all metrics (as in Appendix C.1.2) improve practicality?

**Relation To Broader Scientific Literature:**

The work builds on Swanson et al. (2024) but extends MCTS to multi-objective settings. However, recent Pareto MCTS variants (e.g., Yang et al., 2024a) are not discussed.

**Theoretical Claims:**

The Pareto UCB formula (Equation 8) combines scalar exploration terms with vectorized rewards. This lacks theoretical grounding—how does it guarantee Pareto optimality?

---

> ### Author Rebuttal · Authors · 2025-04-01
>
> Dear reviewer **pmFQ**, thanks for providing valuable feedback to our work! The requested theoretical analysis on PUCB significantly clarifies convergence speed and solution optimality. We address your concerns and questions below and through https://anonymous.4open.science/r/CombiMOTS-0FEB.
>
> ---
> Concerns
> -
> >A) "Table 1 lacks statistical significance."
>
> Following you and **KAF5**, we update Table 1 to improve robustness and relevance. See reply `5` to **KAF5**.
>
> >B) Overfitting concerns - Toxicity prediction
>
> We thank the reviewer for highlighting the small size of ClinTox; we agree models trained on limited data lack real-world accuracy. Our use of ClinTox is intended primarily as a case study due to its clinical relevance (toxicity assays on FDA-approved drugs), despite toxicity data being generally scarce. Nevertheless, ClinTox remains a critical resource for evaluating machine learning models in drug toxicity prediction [1]. We will explicitly underline this limitation in future revisions.
>
> >C) Unfair objective settings across models.
>
> `Appendix C.2` reimplements REINVENT and MARS with our settings (docking score and bioactivity prediction) to investigate alignment of objectives.
> If needed, we will release our implementations of adapted baselines.
> >D) Compared computational cost to baselines.
>
> See reply `2` to **KAF5**.
>
> >E) Broader literature: "Recent Pareto MCTS variants (e.g., Yang et al., 2024a) are not discussed."
>
> Thank you for highlighting this. Our manuscript briefly cites Yang et al. (2024a) (lines 81-82); we further discuss its broader relevance (see reply `1` to **KAF5**).
>
> >F) Page 2: "Fromer & Coley, 2023; Luukkonen & Maagdenberg, 2023" lack full references.
>
> Thanks! We will complete references.
>
> >G) Algorithm 1: Line 16 uses "max(v, selected.P)"—unclear if "max" applies to vectors.
>
> Algorithm 1 omits details for readability. Algorithm 2 (line 17) specifies "max" as element-wise, common in vectorized Pareto MCTS [2].
>
> ---
> Asked Questions
> -
>
> >H) **Theoretical Analysis**: Computational Cost & Pareto-optimal Convergence.
>
> Our manuscript had typos in `Equation 8` and `Algorithm 2` (l.36) on the PUCB formula, omitting a constant of 4. We correct it to:
> $\vec{PUCB(n)} \gets \frac{\vec{R(n)}}{N(n)}+C*\vec{Oracles}\sqrt{\frac{ln(D)+\textbf{4}*ln(1+N(p))}{1+N(n)}}$.
> We provide theoretical grounding by adapting theorems from [ParetoMCTS]. Due to space limits, we summarize key statements below:
> Under assumptions satisfied by our D-dimensional problem setting,
> - **Theorem 1**: Consider a node with a child node $v_k$ visited $T_k(n)$ times in $n$ steps.
> If $v_k$ is a sub-optimal node, $\mathbb{E}(T_k(n))$ is logarithmically bounded.
> $\forall \xi>0, \exists N_0(\xi) \in \mathbb{N^*}, \text{such that}  \ (n \geq N_0(\xi)) \implies \mathbb{E}[T_k(n)] \leq \frac{8ln(n) + 4ln(D)}{(1-\xi)^2 * (\underset{k,d}{min}\Delta_{k,d})^2} + N_0(\xi) + O(1),$ where $(\underset{k,d}{min}\Delta_{k,d})$ is the minimum reward gap (across D dimensions) between $v_k$ and its most dominant node in the Pareto front (i.e. the "farthest away").
> This bound implies: (i) the logarithmic regret $\alpha ln(n)$ implies that sub-optimal node selections decrease over time. (ii) The denominator depends on the minimum reward gap, implying lower regret when sub-optimal nodes are more distinguishable from the Pareto front. (iii) Increasing the number of objectives D also increases regret by a logarithmic factor $\beta ln(D)$, but makes it harder to distinguish sub-optimal and Pareto nodes, directly impacting (ii).
> Thus, (ii) and (iii) motivate us to wisely select our objectives.
>
> - **Theorem 2**: With $I_t$ the selected child node at step $t$ and $(C,\rho$) constants, $\mathbb{P}(I_t$ is sub-optimal$)\leq Ct^{-\frac{\rho}{2}\left(\frac{ \min_{k,d}\Delta_{k,d} }{36}\right)^2} \xrightarrow{t\infty}0.$
> This implies the guaranteed convergence towards optimal nodes at a polynomial rate. In our work, we aim to approach optimal solutions in a reasonable time, thus the need to properly select objectives, efficient oracles and search space size (ablations asked by **6Myy**).
> >I) Why prioritize Docking Scores over QED/SA?
>
> We list our motives below:
> - Fingerprint-based predictions lack structural interpretability (e.g. RationaleRL, MARS, [MolSearch suggested by **KAF5**]) compared to *in-silico* docking simulation.
> - Experiment `C.1.2` suggests that using all objectives leads to worse tradeoffs due to the larger Pareto fronts (now supported by question `H`). An additional experiment (50k rollouts optimizing QED/SA over docking scores) is reported in the repository. We observe that QED slightly increased at the cost of much worse docking scores but the inverse is not true: not considering QED/SA during search still converges towards good scores.
> - As we use Enamine REAL Space, we discuss the non-necessity of SA score (see reply `4` to **KAF5**).
> ---
> Reference
> -
> [1] Tran et al., *JCIM*, 2023
>
> [2] Yang et al., *Commun. Biol.*, 2024

---

### Official Review · Reviewer_h6f3 · 2025-03-14

**Overall Recommendation:** 3

**Summary:**

The paper introduces CombiMOTS, a Pareto Monte Carlo Tree Search (PMCTS) framework for generating dual-target molecules, which are molecules that can interact with two target proteins simultaneously. The authors argue that existing methods often simplify the dual-target optimization problem by linearly combining objectives and neglecting synthetic planning. CombiMOTS addresses these challenges by exploring a synthesizable fragment space and using vectorized optimization constraints to encapsulate target affinity and physicochemical properties. The authors claim that experiments on real-world datasets demonstrate that CombiMOTS generates novel dual-target molecules with high docking scores, enhanced diversity, and balanced pharmacological characteristics. The core idea combines PMCTS with a fragment-based approach and a focus on synthesizability using the Enamine REAL Space.

**Claims And Evidence:**

CombiMOTS is able to perform dual-target optimization. The authors claim that there are intricacies in dual-target optimization that cannot be captured due to simplification by "linear combination of individual objectives", but there are other scalarizing methods (by hypervolume, or distance to utopia point).

The results in Table 1 seem to indicate the metrics used there are not particularly useful, with almost all models achieving scores of >99%. Evaluating generative model quality by validity, uniqueness, novelty, and diversity have been noted as problematic (https://www.sciencedirect.com/science/article/pii/S1740674920300159).

**Essential References Not Discussed:**

- Multiobjective benchmarking for generative models
  - PMO (https://arxiv.org/abs/2206.12411)
- Pareto-optimal generation (https://openreview.net/pdf?id=sWRZxIcR8qK)

Tree-search methods based on fragments and rules based generation have been explored before.
- https://arxiv.org/pdf/2110.06389
- https://proceedings.neurips.cc/paper_files/paper/2020/file/4cc05b35c2f937c5bd9e7d41d3686fff-Paper.pdf
- https://www.nature.com/articles/s42004-024-01133-2

**Experimental Designs Or Analyses:**

The design is logical and sound, however, may be limited in novelty. The authors used a fragment-based tree search to generate synthesizable molecules, and added multi-objective Pareto optimal score, with no significant advancements in either methods. However, I appreciate the non-triviality of implementing this workflow. Additionally, tt is also not clear that CombiMOTs are better than the models presented.

**Methods And Evaluation Criteria:**

It would strengthen the claims of the pareto MCTS being superior if the optimal tradeoffs were quantified, rather than in distribution plots of Figure 3 and 6. A score that quantifies the optimality of the pareto-optimal point? From the plots, it looks like RationaleRL is quite successful as well in the multi-objective task.

**Other Comments Or Suggestions:**

The discussion on pareto optimization can be shortened. The comparison to scalarized results CombiMCTS are important to demonstrate the effect of PUCB optimization, and should be in the main text.

**Other Strengths And Weaknesses:**

Strengths:
- Clear problem definition and motivation.
- Focus on synthesizability is key for molecular generation
- Easy to read and follow

Weakness:
- It is not clear that the claims of CombiMOTs producing dual-target molecules is better than other methods. For example, is it possible to perform a similar case study with a candidate generated by RationaleRL?  A metric to distinguish the pareto-optimal solution being better from CombiMOTS would be beneficial.
- The idea of using synthesizability routes and fragments to give realistic molecules is not new, particular the combination of tree-search methods with retrosynthetic routes (see early response).
- Limited novelty
- Other scalarizing methods that consider pareto-optimization are not considered.

**Questions For Authors:**

Please see comments above.

**Relation To Broader Scientific Literature:**

The work builds upon tree search generation methods for objective oriented molecular generation.

**Theoretical Claims:**

There are no theoretical claims made.

---

> ### Author Rebuttal · Authors · 2025-04-01
>
> Dear reviewer **h6f3**, thank you for your constructive review which improved our perspective! Particularly, your comments provided more clarity regarding the Pareto superiority against baselines. We address your concerns below and through https://anonymous.4open.science/r/CombiMOTS-0FEB.
>
> ---
> Concerns & Replies
> -
> >A) On "linear combination of individual objectives" and other scalarizing methods.
>
> We acknowledge the misuse of “linear combination” as the sole example of scalarization method. We revise our statement by generalizing the advantage of vectorized pareto optimization over scalarized methods – including those mentioned by the reviewer - for the reasons below:
> - Hypervolume assesses the quality of a solution set relative to a reference set: in the dual-target paradigm, baseline sets are either scarce or non-existing, making it hard to use in our task. `Appendix C.3` hints at the capability of CombiMOTS to still effectively find solutions in case of data scarcity.
> - Following your suggestion, we assess each model's ability to discover Pareto optimal molecules by computing their respective first Pareto front and reporting their average R2-distance to utopian point (normalized in range [0 to 2] - lower is better), as well as the size of the Pareto optimal set in parenthesis.
>
> |Model/Task|GSK3B-JNK3|EGFR-MET|PIK3CA-mTOR|
> |-|-|-|-|
> |**CombiMOTS**|0.8075 **(174)**|**0.8419 (268)**|0.8143 **(210)**|
> |**MARS**|**0.7830** (102)|0.8845 (142)|0.8144 (89)|
> |**RationaleRL**|0.8307 (137)|0.9388 (148)|0.8852 (113)|
> |**REINVENT**|0.8223 (93)|0.8518 (177)|**0.8035** (174)|
>
> The main takeaway is that even though all methods present a good overall score, CombiMOTS consistently finds **more Pareto optimal solutions** than others.
> However, this scalarized score doesn't allow identifying inconsistencies among baselines. To clarify this point, we plot radar charts across all metrics used in the manuscript and publish them in the `rebuttal` folder of the anonymous github.
>
> >B) "Selected metrics don't seem useful in Table 1."
>
> Like other visualizations, the role of Table 1 is to highlight inconsistencies of the baselines across metrics (including distribution plots): in all tasks, all baselines perform poorly in at least one metric. Only CombiMOTS performs equally or better than baselines regardless of the task/metric.
>
> About validity: even if only few generations are invalid, it demonstrates a critical flaw in the model not respecting fundamental chemical rules. CombiMOTS use SMARTS-based chemical templates, conformed to real reactions.
>
> As requested by reviewers **pmFQ/KAF5**, we revise Table 1 to be statistically more robust and meaningful. See reply `5` to **KAF5** for details.
> >C) "Selected metrics may be problematic[...]"
>
> We sincerely thank the reviewer for raising the concern about the limitations of using such metrics. In the mentioned paper, we acknowledge that "even the best in silico scores remain fruitless if the suggested compounds cannot be synthesized." Ideally, synthesizing molecules and performing assay experiments would provide the most reliable validation; however, we also agree with reviewer **6Myy** as this typically exceeds the scope of machine learning papers. Thus, like many existing works in this field, we rely on widely-used computational metrics primarily to fairly compare methodologies.
>
> >D) "It looks like RationaleRL is quite successful as well."
>
> We list the weaknesses of RationaleRL in our experiments:
> - GSK3B-JNK3: Low uniqueness and lowest diversity;
> - EGFR-MET: Low diversity, weak binding affinity with MET (Fig.6A) and high SA score;
> - PIK3CA-mTOR: Low uniqueness, novelty, diversity and fairly low QED score (Fig.6B).
>
> We discuss the low diversity of RationaleRL. Its workflow revolves around (i) extracting and merging "rationales" from high-property compounds, (ii) training & finetuning a graph completion module to obtain novel molecules from the merged rationales. Generation occurs by autoregressively complete the same structural cores, thus leading to poor diversity.
> We plot and add (to the repository) t-SNE figures of generated compounds from all models: RationaleRL generations are clustered. In drug design, RationaleRL is better suited for Lead Optimization rather than Hit Discovery.
>
> >E) Suggested References
>
> We thank the reviewer for suggesting related material. We will integrate similar topics to our related work. In the meanwhile, we comment on how our work differs:
>
> Multiobjective JANUS focuses on molecular optimization, not de novo design. [1]
>
> SynNet [2] and MEEA* [3] tackle synthesis planning, which answers: "How can we synthesize this molecule given available stock?"
> This is the inverse of our approach, which asks: "Given available stock, what optimal products can be created?" See reply `4` to **KAF5** for more about our motives.
>
> ---
> Reference
> -
> [1] Kusanda et al., *NeurIPS'22*, 2022
>
> [2] Gao et al., *arXiv*, 2021
>
> [3] Zhao et al., *Commun. Chem.*, 2024

---

> > ### Comment · Reviewer_h6f3 · 2025-04-05
> >
> > Thank you for the author rebuttal.
> >
> > A) Thank you for the work in producing these results.
> >
> > B/C) I am not requesting the synthesis or experimentation of any chemical compounds (I agree this would be very out of scope for an ML conference). However, the authors should add a statement about the pitfalls of using metrics such as validity, uniqueness, novelty, and diversity. Such metrics can be easily hacked, for example, by adding a couple "C"s into the smiles string: [AddCarbon model] (https://www.sciencedirect.com/science/article/pii/S1740674920300159)
> >
> > Thank you for the additional work. I will increase my score.

---

> > > ### Author Response · Authors · 2025-04-07
> > >
> > > Dear reviewer **h6f3**, we reiterate our thanks for your valuable advice and perspective on the used metrics, and for your constructive review overall on Pareto superiority !
> > >
> > > We will make sure to discuss the pitfalls of such commonly used metrics in future versions. We share your opinion as this tackles broader applications in drug discovery - not only limited to our work - that should also be addressed in future works.
> > >
> > > We thank you again for your thoughtful feedback during the entire review process.
> > >
> > > We also thank you for increasing your score!

---

### Official Review · Reviewer_6Myy · 2025-03-14

**Overall Recommendation:** 3

**Summary:**

This paper introduces CombiMOTS, a novel method for dual-target molecule generation.  It addresses the limitations of existing approaches, which often simplify the multi-objective nature of the problem into a linear combination of objectives and may not consider synthesizability.  CombiMOTS leverages Pareto Monte Carlo Tree Search (PMCTS) within a fragment-based drug discovery framework.

**Claims And Evidence:**

The claims made in the submission are generally well-supported by the evidence provided.

*   **Claim:** CombiMOTS outperforms existing methods in generating dual-target molecules with better trade-offs between target engagement and molecular properties.
    *   **Evidence:**  Table 1 and Figures 3, 6, and the supplementary figures provide strong quantitative evidence.  CombiMOTS consistently achieves higher novelty and diversity scores, while maintaining competitive or superior docking scores compared to the baselines. The density plots visually demonstrate the superior trade-offs.
    *   **Concerns:** None major.

*   **Claim:** The use of Pareto optimization is crucial for achieving these improved trade-offs.
    *   **Evidence:**  The comparison with the scalarized version of CombiMOTS (Figure 7 and Appendix C.1.1) provides compelling evidence.  The scalarized version performs worse in terms of finding balanced properties, highlighting the importance of the Pareto approach.
    *   **Concerns:** None major.

*   **Claim:** The fragment-based approach and use of the Enamine REAL Space ensure synthesizability.
    *   **Evidence:**  The methodology section clearly describes the process of mapping fragments to REAL Space building blocks and using reaction templates. The reported SA scores are generally low, indicating good synthesizability. The comparison with Swanson et al. (2024), who focused on the REAL space.
    *   **Concerns:** While SA score is a good proxy, it's not a perfect guarantee of synthesizability. Experimental validation would *strengthen* this claim, but is beyond the scope of a typical ML paper.

* **Claim:** CombiMOTS has practical utility, analysis of Case Studies
    * **Evidence:** Analysis of case studies for compounds generated for GSK3B-JNK3.
    * **Concerns:** None.

**Essential References Not Discussed:**

Essential related works have been discussed.

**Experimental Designs Or Analyses:**

The experimental designs are generally sound.

*   **Datasets:**  Using established datasets (GSK3β-JNK3) and curating new ones (EGFR-MET, PIK3CA-mTOR) according to established protocols is good practice.  The data curation process is described in detail in the appendices.
*   **Baselines:**  The chosen baselines are relevant and represent different approaches to molecule generation.  The authors made efforts to reproduce the baselines in their best reported settings and even adapted them to use docking scores as objectives for a fairer comparison.
*   **Metrics:**  The use of multiple metrics to evaluate different aspects of the generated molecules is comprehensive.
*   **Ablation Studies:**  The comparisons with the scalarized version of CombiMOTS and the six-objective version (in the appendix) are valuable ablation studies that demonstrate the importance of the Pareto approach and provide insights into the impact of adding more objectives.
*    **Concerns:** Ablation study of the Fragment Extraction and Search Space Reduction is missing.

**Methods And Evaluation Criteria:**

The proposed methods and evaluation criteria are generally appropriate for the problem.

*   **Methods:** The use of PMCTS is a well-established technique for search and optimization.  Adapting it to a combinatorial, multi-objective, and fragment-based setting for drug discovery is novel and well-motivated.  The use of the Enamine REAL Space is a good choice for ensuring synthesizability.
*   **Evaluation Criteria:**  The chosen metrics (validity, uniqueness, novelty, diversity, docking score, QED, SA) are standard and relevant for evaluating generative models in drug discovery.  Using multiple target pairs (including newly curated datasets) strengthens the evaluation.  Comparing against relevant baselines (RationaleRL, REINVENT, MARS) is also appropriate.
*    **Concerns** The justification for choosing to prioritize the molecular docking score, other than the other metrics, is not solid.

**Other Comments Or Suggestions:**

Reference to the **Weaknesses** part.

**Other Strengths And Weaknesses:**

**Strengths:**

*   **Novelty:** The combination of PMCTS with a combinatorial, fragment-based approach for dual-target molecule generation is novel and well-motivated.
*   **Effectiveness:** The experimental results demonstrate that CombiMOTS outperforms existing methods on the chosen tasks.
*   **Synthesizability:** The focus on synthesizability through the use of the Enamine REAL Space is a significant strength.
*   **Comprehensive Evaluation:**  The use of multiple metrics and target pairs, along with ablation studies, provides a thorough evaluation.

**Weaknesses:**

*   **Computational Cost:** The paper acknowledges the computational cost of PMCTS, particularly with more objectives.  While the authors provide runtimes, a more detailed discussion of scalability and potential optimizations could be helpful.
* **Justification of Docking score:** The paper needs more justification, why molecular docking score is the main objective.
*   **Limited Scope:** The evaluation is limited to three dual-target pairs and a specific set of objectives.  While this is understandable, it would be interesting to see how CombiMOTS performs on a wider range of tasks and with different objective functions.
* **Missing Ablation study**: The ablation study is missing for Fragment Extraction and Search Space reduction steps.

**Questions For Authors:**

1.  **Computational Cost and Scalability:** Can you elaborate on the scalability of CombiMOTS, particularly as the number of objectives or the size of the search space increases?  Are there any specific optimizations you have considered or plan to explore to improve efficiency? How does the computational cost compare to the baselines (even if approximate)?
2.  **Objective Function Design:** You prioritize docking scores over QED and SA.  Could you elaborate on the rationale behind this choice?  What are the potential limitations of this prioritization, and how might it affect the generated molecules? Have you experimented with different weightings or combinations of objectives, and if so, what were the results?
3.  **Generalizability:** How do you envision CombiMOTS being applied to other drug discovery tasks beyond dual-target inhibitor design?  Could it be adapted to handle different types of targets (e.g., protein-protein interactions) or different objective functions?
4.  **Fragment Extraction:** Could you elaborate on the choice of FGIB for fragment extraction? Were other fragment extraction methods considered, and if so, why was FGIB chosen? What are the potential limitations of the chosen fragment extraction method?
5. **Search Space Reduction.** How sensitive is the performance of CombiMOTS to the Tanimoto similarity threshold used for search space reduction? Did you perform any experiments to optimize this threshold?
6. **Ablation Study**: The ablation study of the Fragment Extraction and Search Space Reduction steps, are important to show the efficiency of the whole pipeline.

**Relation To Broader Scientific Literature:**

The paper is well-situated within the broader scientific literature on drug discovery, generative models, and multi-objective optimization. It builds upon:

*   **Fragment-Based Drug Discovery (FBDD):**  The paper clearly positions itself within the FBDD paradigm and cites relevant work.
*   **Generative Models for Molecules:**  It cites and compares against relevant generative models, including VAE-based, RL-based, and MCMC-based approaches.
*   **Multi-Objective Optimization:**  It correctly references Pareto optimization and its application in other fields.
*   **Monte Carlo Tree Search (MCTS):**  It builds upon existing work on MCTS and PMCTS, citing relevant papers.

**Theoretical Claims:**

The paper does not present significant theoretical claims in the form of theorems and proofs.  The theoretical underpinning is the application of Pareto optimality and MCTS, which are well-established concepts. The description of Pareto dominance and Pareto fronts is correct.

---

> ### Author Rebuttal · Authors · 2025-04-01
>
> Dear reviewer **6Myy**, we appreciate your insights which helped improve our work! Particularly, ablation studies allowed us to strengthen our claims. We address your concerns below and through https://anonymous.4open.science/r/CombiMOTS-0FEB.
>
> ---
> Concern
> -
> > A) "SA score doesn't guarantee perfect synthesizability".
>
> See reply `4` to **KAF5**.
>
> ---
> Asked Questions
> -
> > 1. **Computational Cost and Scalability** - *#Objectives*, *Runtime* & *Solution Convergence*.
> > 2. **Objective Function Design** - Docking Score over QED/SA.
>
> &rarr; See reply `I` to **pmFQ** and `2` to **KAF5**.
> > Have you experimented with different weightings[...]?
>
> To clarify the use of weights, CombiMOTS uses property vectors to compute Pareto fronts and sample nodes: weights don't affect the relative difference between nodes.
>
> > 4. **Fragment Extraction** - FGIB & 6. **Ablation Study**
>
> FGIB extends the widely used BRICS [1] method by breaking molecules into retrosynthetically interesting fragments, which is preferred over rule/frequency-based methods when combined with Enamine REAL Space.
> Some works adapt BRICS to needs with a fixed goal (e.g. connectivity-awareness [MiCaM [2]], ADMET explanability [pBRICS [3]]), thus not suited for the dual-inhibition task where target proteins are user-defined.
> Our goal is to capture high-property fragments for any **target property**.
> To further justify our choice, we report results from 10,000 rollouts on the GSK3B-JNK3 task when replacing FGIB with naive BRICS and MiCaM, a connection-aware motif-mining approach to decompose known active compounds.
> |Method|#BuildingBlocks|#PossibleProducts|#Dual-Actives|
> |-|-|-|-|
> |BRICS|4,430 (747 not found by FGIB)|~1.7M|1,815/12,559 (**14.45%**)|
> |MiCaM|12,274 (8,428 not found by FGIB)|~26M|1,445/13,263 (**10.90%**)|
> |FGIB (base)|14,366 (10,683 unique vs. BRICS, 10,520 vs. MiCaM)|~25M|3,662/15,423 (**23.74%**)|
>
> FGIB successfully identified goal-specific fragments, resulting in a **significantly higher rate** of "dual-active" compounds predicted to interact with both proteins.
>
> We report metrics/distribution plots in the `rebuttal` folder of the anonymous github.
> All methods exhibit similar performance, which is sound as FGIB only impacts the search space through the selection of **initial blocks**: MCTS objectives and convergence speed are "as good", but the attainable space contains less dual actives.
> > B) Potential limitations of FGIB.
>
> FGIB uses MLP-based modules to learn data features: its performance depends on data quality. However, our ablation study (`Appendix C.3`) demonstrates that combined with Pareto MCTS, data scarcity is alleviated to some extent.
>
> > 5. **Search Space Reduction** - Thresholds & 6. **Ablation Study** - Search Space Reduction
>
> We investigate the use of various thresholds and report results from 10k rollouts on the GSK3B-JNK3 task for lower and higher values of 0.3, 0.5 & 0.6.
> |Threshold|# Blocks|# Possible Products|# Dual-Actives|
> |-|-|-|-|
> |0 (Full Space)|139,493|>31B|(not done)|
> |0.3|54,811|~478M|2,833/13,556 **(20.90%)**|
> |0.4 (base)|14,366|~25M|3,662/15,423 **(23.74%)**|
> |0.5|3,737|~1.1M|1,380/11,632 **(11.86%)**|
> |0.6|858|~43K|317/9433 **(3.36%)**|
>
> You may refer to metrics and figures added to the repository.
> We make three observations:
>
> - The threshold greatly affects the search space size due to the small nature of FGIB fragments and Enamine blocks. Tanimoto metric being fingerprint-based, changing a small motif is relatively impactful.
> - For higher thresholds, the search space is too small to allow good exploration. There is a significant drop in #dual-actives, and metrics also show that using thresholds above 0.6 yield a drop in diversity (88.67→78.73%) and consistency across molecular properties.
> - For lower thresholds, CombiMOTS has "more room to explore" but finds worse tradeoffs across QED and #dual-actives. As more reactions are possible, Pareto fronts are larger and convergence to optimal solutions is slower (see `H) Theroretical Analysis` to reviewer **pmFQ**).
>
> Optimal thresholds are specific to the target properties, but our experiments suggest that a search space of magnitude 10M yields better convergence within a reasonable budget (~10k rollouts).
>
> > 3. Generalizability
>
> While dual-target inhibition is our main task, we design CombiMOTS can be adapted by customizing property oracles as objectives, or by targeting specific biological systems, narrowing the search space on fragments of interest.
>
> Among applications, reviewer **KAF5** and yourself mentioned:
> - Protein-Protein Interaction (PPI) modulator discovery, where search can aim to uncover molecules tailored to PPI interface [Hot2Mol [4], GENiPPI [5]].
> - Toxicity Optimization (see reply `6` to **KAF5**).
> - Molecular Selectivity (see discussion `B` with **KAF5**).
>
> ---
> Reference
> -
> [1] Degen et al., *ChemMedChem*, 2008
>
> [2] Geng et al., *arXiv*, 2023
>
> [3] Vangala et al., *JCIM*, 2023
>
> [4] Sun et al., *bioRxiv*, 2024
>
> [5] Wang et al., *J. Cheminform.*, 2024

---

> > ### Comment · Reviewer_6Myy · 2025-04-08
> >
> > Thank you for your detailed rebuttal and for conducting the additional experiments regarding fragment extraction and search space reduction thresholds.
> > I appreciate the effort to provide these ablation studies (addressing Questions 4, 5, 6). They offer helpful clarification on the impact of FGIB compared to other methods and the sensitivity to the Tanimoto threshold, providing further insight into the pipeline's design.
> > However, while these clarifications are welcome, my overall assessment of the paper remains consistent with my initial review. The concerns regarding the justification for prioritizing docking scores (Question 2), the potential computational cost and scalability (Question 1), and the relatively limited scope evaluated still weigh significantly in my view.
> > Therefore, although the additional experiments are informative, they do not sufficiently overcome the previously identified weaknesses to warrant an increase in the score at this time.
> > My recommendation remains unchanged.

---

> > > ### Author Response · Authors · 2025-04-09
> > >
> > > Dear reviewer **6Myy**, we sincerely thank you for the deep understanding of our work throughout the review process. Your insights enabled to provide further analysis on important ablation studies and practical aspects of our method.
> > >
> > > Within the rebuttal period, we did our best attempting to address your questions on objective design and scalability with theoretical analysis and additional experiments: we try to further elaborate on these points.
> > >
> > > - The theoretical analysis to **pmFQ** suggests than intuitively, using more objectives leads to slower convergence because hard to distinguish Pareto optimal molecules. We support this with more evidence by showing below the size of the first six Pareto fronts on 10K samples (generated by CombiMOTS on the GSK3B-JNK3 task), as well as the number of Pareto fronts.
> > >
> > > |Pareto Rank/Objective Setting|Docking+Activity (4 Obj)|Activity+QED+SA (4 Obj)|All Six Objectives|
> > > |-|-|-|-|
> > > |Rank 1|60|69|729|
> > > |Rank 2|135|204|1416|
> > > |Rank 3|180|257|1809|
> > > |Rank 4|233|370|1771|
> > > |Rank 5|263|456|1456|
> > > |Rank 6|316|502|1153|
> > > |Number of fronts|40|29|12|
> > >
> > > We observe that as expected, using six objectives leads to fewer fronts which are individually more populated.
> > >
> > >
> > > - Docking score is critical in drug design because it directly evaluates ligand-target binding affinity, which is essential for therapeutic efficacy. Unlike QED/SA, which focus on general drug-likeness or synthetic feasibility, docking scores are structurally and energetically tied to the biological activity of the molecule. Studies have shown that incorporating docking scores during molecular generation leads to higher binding affinity and improved hit identification [1], whereas QED/SA cannot ensure target specificity [2, 3]. Therefore, docking score should be prioritized in early drug discovery stages, with QED and SA used later for optimization [4].
> > > - This is supported by our additional experiment during rebuttal (comparing both settings). We observed that not optimizing docking scores leads to a significant decrease of quality, but not optimizing QED/SA still leads to acceptable scores, as well as better docking scores. The table above also supports this, as using docking scores over QED/SA interestingly allows to better differentiate Pareto fronts. These results are empirical but align perfectly with the necessity to consider both target affinity and molecular properties within the limitations of the objective settings for Pareto MCTS.
> > >
> > >
> > >
> > > Regarding the scope of our work we would like to remind that, though we agree the submitted manuscript did not discuss on other applications, we purposefully focused on the topic of dual-target molecules to fit within the format of a conference paper.
> > > The additional case studies conducted after you and reviewer's **KAF5** suggestions [Toxicity Optimization, Molecule Selectivity] exemplify how CombiMOTS could adapt to any multi-objective optimization setting.
> > >
> > > Besides the limits inherent to the rebuttal, we also deem that extensively elaborating such applications in a single paper could confuse readers.
> > > However, we agree that discussing our method's generalizability is important: we intend to integrate these results in future versions' appendices.
> > >
> > > We thank you again for the thoughtful comments and the time invested!
> > >
> > > ---
> > > Reference
> > > -
> > > [1] Agu et al., *Scientific Reports*, 2023
> > >
> > > [2] Xu et al., *F1000Research*, 2024
> > >
> > > [3] Chenthamarakshan et al., *Advances in Neural Information Processing Systems 33*, 2020
> > >
> > > [4] Xue et al., *Bioinformatics*, 2025

---

### Decision · Program_Chairs · 2025-05-01

**Decision:**

Accept (poster)

**Comment:**

This paper proposes CombiMOTS: a method utilizing Pareto MCTS to search for molecules simultaneously optimizing affinity towards two targets. To ground the generations in a synthetically accessible space, the authors project target-specific fragments to available building blocks, and then use those to assemble final molecules.

After a fruitful rebuttal period that lead to several score increases, all reviewers were voting to accept this paper, although weakly so. On the positive side, reviewers enjoyed the good quantitative performance of CombiMOTS, as well as its solid motivation and practical design. On the negative side, reviewers asked for further ablations, statistical analyses, and discussion of theoretical underpinnings, all of which were subsequently discussed by the authors. Some reviewers also noted somewhat limited novelty, with CombiMOTS mostly being a combination of pre-existing techniques; however, this point is itself of limited importance in the applications track, where practicality is the main concern.

After reviewing all available information and reading the paper in detail, I believe this manuscript is a decent contribution to ICML's program. Although not highly innovative, CombiMOTS could be interesting to practitioners. Therefore, I (weakly) recommend acceptance of this paper. My main reason for not going for a stronger recommendation is that CombiMOTS may not be as widely applicable as some other methods, thus perhaps interesting to a slightly narrower audience than an average ICML-accepted paper in the field of AI for Science. Finally, I would encourage the authors to incorporate reviewer feedback into the final version of their paper, and spend some extra time polishing the text and LaTeX notation; having read the work myself I believe it would be nice to make the reading experience a bit smoother.